# Multimodal Few-Shot Point Cloud Segmentation via Agent Adaptation and Discriminative Deconfusion

## Abstract

Few-shot 3D point cloud segmentation (FS-PCS) aims to leverage a limited amount of annotated data to enable the segmentation of novel categories. Most existing studies rely on single-modal point cloud data and have not fully explored the potential of multimodal information. In this paper, we propose a novel FS-PCS framework, Multimodal Agent Adaptation and Discriminative Deconfusion (MAD). MAD incorporates three modalities: images, point clouds, and category text embeddings. To fuse multimodal information, we propose the Multimodal Semantic Agents Correlation Aggregation (M-SACA) module, which fuses multimodal features through agent-level correlation and uses text affinity for category semantic learning. To alleviate semantic gaps between the support set and query set in multimodal features, we propose the Semantic Agents Prototypes Adaptation (SAPA) module, which generates multimodal agents for query and support sets, adjusting prototypes to adapt the query feature space. To alleviate intra-class confusion, we introduce the Discriminative Deconfusion (DD) module, which preserves intra-class consistency through residual adapters and generator weights. Experiments on the S3DIS and ScanNet datasets demonstrate that MAD attains state-of-the-art performance, improving mIoU by 3%–7%. Our method can significantly improve segmentation results and suggest valuable insights for future studies. The code will be publicly available.

## 1 Introduction

Recently, 3D computer vision has garnered increasing attention Chen et al. (2024). As a crucial branch, 3D point cloud semantic segmentation plays a pivotal role by providing semantic understanding essential for foundational applications Xiao et al. (2024); Zhan et al. (2025); Jiang et al. (2024). Fully supervised 3D point cloud semantic segmentation (FS-PCS) Qi et al. (2017a;b); Thomas et al. (2019); Zhao et al. (2021a); Wu et al. (2022); Zhan et al. (2023); Wu et al. (2024) has advanced rapidly in recent years, achieving remarkable success.

Due to the substantial annotation costs associated with FS-PCS, further exploration remains limited. This constraint is particularly pronounced when expanding to new categories, which typically requires retraining or fine-tuning. To address this challenge, Few-Shot 3D point cloud segmentation (FS-PCS) Zhao et al. (2021b); Zhang et al. (2023); Zhu et al. (2024) has been established. The critical issue in FS-PCS lies in effectively transferring semantic information from the support point cloud to the query point cloud. Based on attMPTI Zhao et al. (2021b), several methods Ning et al. (2023); He et al. (2023); Wang et al. (2025) have been developed to tackle this task. However, the attMPTI benchmark suffers from foreground leakage and sparse point distributions. To address these limitations, COSeg An et al. (2024) introduced a more stringent benchmark that better reflects real-world 3D point cloud scenarios. It remains highly challenging due to the complexity of dense point cloud scenes.

In few-shot image semantic segmentation, pre-training on large-scale datasets such as ImageNet Deng et al. (2009) has enabled the extraction of highly expressive feature representations, leading to impressive segmentation performance. Image data has great potential to enhance the understanding of FS-PCS. Motivated by this Quiroga et al. (2005); Nanay (2018); An et al. (2025b), leveraging

Figure 1: Comparison of previous multimodal FS-PCS method (Right) with our MAD (Left).

existing pre-trained text and image models to enhance FS-PCS represents a promising research direction.

We identified three issues in the current multimodal FS-PCS setup: **(1)** The existing method An et al. (2025b) typically only fuses multimodal correlation features of the query point cloud and then employs a multi-layer perceptron (MLP) to predict the category of each point. Without generating the multimodal representation of the supporting point cloud, this approach lacks a unified query and support feature format, which prevents it from capturing the dependency correlation between query and support sets, thereby hindering the model's ability to identify new categories under limited supervision. **(2)** Since the query and support sets differ in visual appearance and geometric structure, the multimodal correlation features may have semantic gaps. Without an effective adjustment mechanism, these discrepancies can significantly degrade the quality of the generated prototype, resulting in reduced segmentation performance. **(3)** The experiment found that the prediction had intra-class confusion. Although text embedding provided class-level prior knowledge, simple fusion would lead to intra-class ambiguity if it were not effectively aligned with visual concepts.

To address these challenges, we propose a novel framework named Multimodal Agent Adaptation and Discriminative Deconfusion (MAD). The framework enhances cross-modal correlation modeling capabilities, adapts the query set semantic feature space, and maintains intra-class consistency, thereby achieving high-quality multimodal FS-PCS. As shown in Figure 1, unlike current work, we propose fusing query and support correlation feature simultaneously, regenerating adaptive prototypes, utilizing cosine similarity for preliminary predictions, and ultimately removing discriminative confusion through text affinity. Overall, our main contributions:

**(i)** We propose a novel architecture that fuses multimodal features by aggregating semantic correlations between the query and support sets. To the best of our knowledge, this is the first study to explore the simultaneous fusion of query and support point cloud multimodal correlation features in traditional FS-PCS. **(ii)** We propose a new multimodal FS-PCS model, MAD, to effectively alleviate the issues of multimodal fusion, which includes Multimodal Semantic Agents Correlation Aggregation (M-SACA), Semantic Agents Prototypes Adaptation (SAPA), and Discriminative Deconfusion (DD). **(iii)** Extensive experiments have verified that the MAD is the state-of-the-art method. It can inspire future research.

## 2 RELATED WORK

### 2.1 3D POINT CLOUD SEGMENTATION

In recent years, point cloud segmentation (PCS) Landrieu & Simonovsky (2018); Zhang et al. (2021); Xiao et al. (2022); Li et al. (2023); Zhan et al. (2023); Han et al. (2024); Kolodiazhnyi et al. (2024) has advanced rapidly due to deep learning. PointNet Qi et al. (2017a) is a pioneering work that utilizes the permutation invariance of max pooling to extract features from point clouds. To extract local features and efficiency, PointNet++ Qi et al. (2017b) uses the radius ball method to determine local domains and FPS downsampling. KPConv Thomas et al. (2019) proposes dynamic kernel aggregation of local point cloud features, which requires setting the kernel size according to the scene. DGCNN Wang et al. (2019) proposes edge convolution and then uses the updated features of dynamic graphs. Large-scale PCS tasks attract a large number of researchers due to their complexity. RandLA-Net Hu et al. (2020) introduces random sampling and retains important features by expanding the receptive field. Since PCT Guo et al. (2021) and Point Transformer Zhao et al. (2021a) introduced the use of Transformers for processing point clouds, numerous Transformer-based methods Lai et al. (2022); Wu et al. (2022; 2024) have been proposed, significantly advancing

research in point cloud semantic segmentation. However, these dependencies are fully annotated and can not perform semantic segmentation of new categories.

## 2.2 FEW-SHOT 3D POINT CLOUD SEGMENTATION

The attMPTI Zhao et al. (2021b) is the pioneering work of FS-PCS that performs label propagation from the support to the query set by constructing the KNN graph. Building on this, subsequent works Ning et al. (2023); He et al. (2023) have focused on extracting robust prototypes and reducing semantic bias from the query set. To retain the ability to split base classes, GWs Xu et al. (2023) proposed geometric word components shared by base classes and new classes. Meanwhile, DLE Li et al. (2024) first locates high-confidence areas and then expands the area through similarity. TaylorSeg Wang et al. (2025) transforms the problem of representing the local structure of irregular point clouds into a polynomial fitting problem, which can effectively reduce the difference in feature distribution between the support and query sets. To alleviate the limitations of sparse knowledge, GFS-VL An et al. (2025a) introduces 3D VLMs, which improve learning through pseudo-label filtering and a base mixing strategy. However, the benchmark suffers from prospect leakage and sparse point distribution problems. Therefore, COSeg An et al. (2024) established a new benchmark and proposed optimizing class-specific multi-prototypical correlation (CMC) to calculate the features of each query point. MM-FSS An et al. (2025b) introduces multimodal data based on COSeg and proposes MSF to integrate multimodal data, verifying the potential of multimodal data. How to effectively fuse multimodal data remains an open question that requires further investigation.

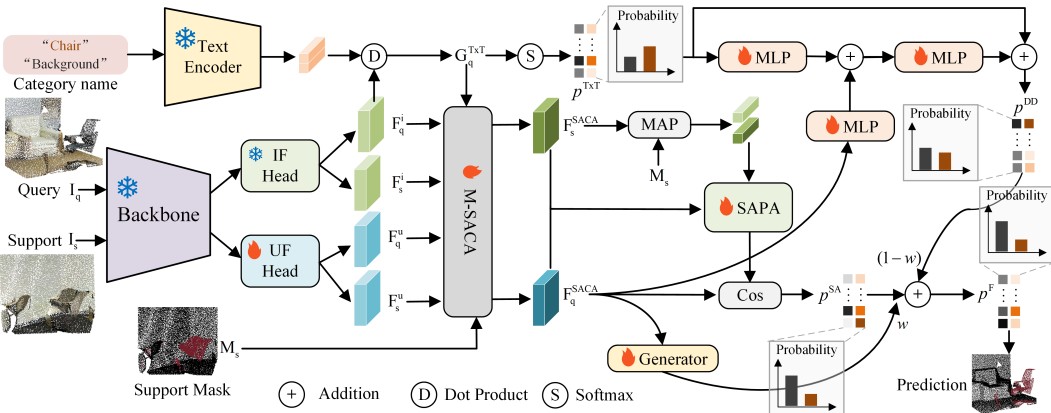

Figure 2: The architecture of the MAD model. Given the query and support point clouds, we generated correlation features $\mathbf{F}_{q/s}^{i/u}$ by IF Head and UF Head, and intra-class correlation features $\mathbf{F}_{q/s}^{SACA}$ are extracted through the M-SACA module. Furthermore, we use the SAPA module to adaptively adjust prototypes by aligning the feature spaces of the query and support sets. Finally, text embedding and $\mathbf{F}_q^i$ affinity provide class-level prior knowledge, which is adapted through the residual MLP to remove category discriminative ambiguity. The model is displayed in 1-way 1-shot.

## 3 METHODOLOGY

Our approach aims to explore the effective fusion of multimodal data in FS-PCS, providing a reference for other fields.

## 3.1 PROBLEM DEFINITION

FS-PCS aims to enable the model to segment new categories, i.e., the training set $\mathcal{D}_{\text{train}}$ and the test set $\mathcal{D}_{\text{test}}$ do not have the same categories ($\mathcal{C}_{\text{train}} \cap \mathcal{C}_{\text{test}} = \emptyset$). Following prior work Vinyals et al. (2016), the problem is organized into a series of sub-tasks, known as episodes. Each episode is formulated as an $N$-way $K$-shot semantic segmentation task. Specifically, the support set is defined as $\mathcal{S} = \left\{ \left\{ \left( I_s^{n,k}, M_s^{n,k} \right) \right\}_{k=1}^{K} \right\}_{n=1}^{N}$, and the query set can be represented as $\mathcal{Q} = \left\{ \left( I_q^n, M_q^n \right) \right\}_{n=1}^{N}$.

$I$ is the input point cloud, and $M$ is the mask. FS-PCS can be described as utilizing information from the support set $\mathcal{S}$ to classify the query samples $\left\{I_q^n\right\}_{n=1}^N$ into $N+1$ categories, including a background class.

Multimodal FS-PCS introduces category names and image forms. Category names are encoded as text embeddings using the text encoder in LSeg Li et al. (2022) and CLIP Radford et al. (2021), while image features are extracted through its visual encoder. For convenience, unless otherwise specified, the subsequent sections will be under the 1-way 1-shot setting. Thus, the support set and query set are represented as $\mathcal{S} = (\mathbf{I}_s, \mathbf{M}_s)$ and $\mathcal{Q} = (\mathbf{I}_q, \mathbf{M}_q)$ respectively.

### 3.2 OVERVIEW

As shown in Figure 2, support and query point clouds are input separately to the backbone. The Intermodal Feature (IF) Head and Unimodal Feature (UF) Head are then used to extract features from two distinct modalities. Subsequently, the M-SACA module performs modality fusion to generate the correlation features. Adapt the prototype through the SAPA module, and use query correlation features and the adjusted prototype to achieve preliminary segmentation. In addition, category names are transformed into text embeddings using the text encoder. The affinity between text embeddings and $\mathbf{F}_q^i$ can be used to make prior predictions. Finally, to mitigate the issue of discriminative confusion, a residual adapter structure is employed to reduce ambiguity in probability predictions and subsequently fuses the dynamically generated weights with visual prediction.

**Text Embedding:** The text encoder $\mathcal{E}_{\text{TxT}}$ from LSeg is used to encode semantic category text $\mathcal{C}$. That is, $\mathbf{T} = \mathcal{E}_{\text{TxT}}(\mathcal{C})$, yielding the text embedding. $\mathbf{T} = [\mathbf{t}_0; \mathbf{t}_1; ...; \mathbf{t}_N] \in \mathbb{R}^{N_C \times D_t}$, where $D_t$ is the text embedding feature dimension and $N_C = N + 1$, which conforms to the $N$-way setting. $\mathbf{t}_0$ represents the background text embedding, and the rest are target category embeddings.

**Visual Embedding:** The original point cloud only has low-level features such as 3D coordinates or color information. Therefore, the support point cloud $\mathbf{I}_s$ and query point cloud $\mathbf{I}_q$ are input into the shared backbone network $\mathcal{E}_{\text{Vis}}$ to obtain the encoded high-dimensional features $\mathbf{F}_s = \mathcal{E}_{\text{Vis}}(\mathbf{I}_s)$ and $\mathbf{F}_q = \mathcal{E}_{\text{Vis}}(\mathbf{I}_q)$. Then, using IF Head ($\mathcal{H}_{\text{IF}}$) and UF Head ($\mathcal{H}_{\text{UF}}$), we obtain the IF/UF features for support/query, defined as:

$$\begin{cases} \mathbf{F}_s^i = \mathcal{H}_{\text{IF}}(\mathbf{F}_s), & \mathbf{F}_s^u = \mathcal{H}_{\text{UF}}(\mathbf{F}_s), \\ \mathbf{F}_q^i = \mathcal{H}_{\text{IF}}(\mathbf{F}_q), & \mathbf{F}_q^u = \mathcal{H}_{\text{UF}}(\mathbf{F}_q), \end{cases} \tag{1}$$

where $\mathbf{F}_s^u \in \mathbb{R}^{N_s \times D}$, $\mathbf{F}_q^u \in \mathbb{R}^{N_q \times D}$, and $D$ is the dimension of the UF feature. To align with text embedding, the dimensions of the IF feature are kept consistent with those in LSeg Li et al. (2022) and CLIP Radford et al. (2021). Hence, $\mathbf{F}_s^i \in \mathbb{R}^{N_s \times D_t}$, $\mathbf{F}_q^i \in \mathbb{R}^{N_q \times D_t}$. $N_s$ and $N_q$ are the number of points in the support and query sets, respectively. Note that under the COSeg settings, generally $N_s \neq N_q$.

### 3.3 MULTIMODAL SEMANTIC AGENTS CORRELATION AGGREGATION (M-SACA)

As shown in Figure 3 (Left), the foreground/background semantic agents are generate by agent clustering $\mathcal{F}_{\text{agent}}$. First, the sampling points are determined using the 3D coordinates $\mathbf{X}_s$ through farthest point sampling (FPS), retaining the main structures of the point cloud scene. Then, by clustering other point features to the sampling points, agent point features are formed. The specific definition is as follows:

$$\begin{cases} \mathbf{A}_{\text{fg}}^i = \mathcal{F}_{\text{agent}}(\mathbf{F}_s^i, \mathbf{M}_s, \mathbf{X}_s), & \mathbf{A}_{\text{bg}}^i = \mathcal{F}_{\text{agent}}(\mathbf{F}_s^i, \tilde{\mathbf{M}}_s, \mathbf{X}_s), \\ \mathbf{A}_{\text{fg}}^u = \mathcal{F}_{\text{agent}}(\mathbf{F}_s^u, \mathbf{M}_s, \mathbf{X}_s), & \mathbf{A}_{\text{bg}}^u = \mathcal{F}_{\text{agent}}(\mathbf{F}_s^u, \tilde{\mathbf{M}}_s, \mathbf{X}_s), \end{cases} \tag{2}$$

where $\mathbf{A}_{\text{fg}}^i, \mathbf{A}_{\text{bg}}^i \in \mathbb{R}^{N_A \times D_t}$, $\mathbf{A}_{\text{fg}}^u, \mathbf{A}_{\text{bg}}^u \in \mathbb{R}^{N_A \times D}$, and $\tilde{\mathbf{M}}_s$ is the inverse mask of $\mathbf{M}_s$. $N_A$ is the number of agent points. $\mathbf{X}_s$ is the 3D coordinates of the points. $\mathbf{A}_{agent}^i = Concat(\mathbf{A}_{\text{fg}}^i, \mathbf{A}_{\text{bg}}^i) \in \mathbb{R}^{N_C \times N_A \times D_t}$ and $\mathbf{A}_{agent}^u = Concat(\mathbf{A}_{\text{fg}}^u, \mathbf{A}_{\text{bg}}^u) \in \mathbb{R}^{N_C \times N_A \times D}$.

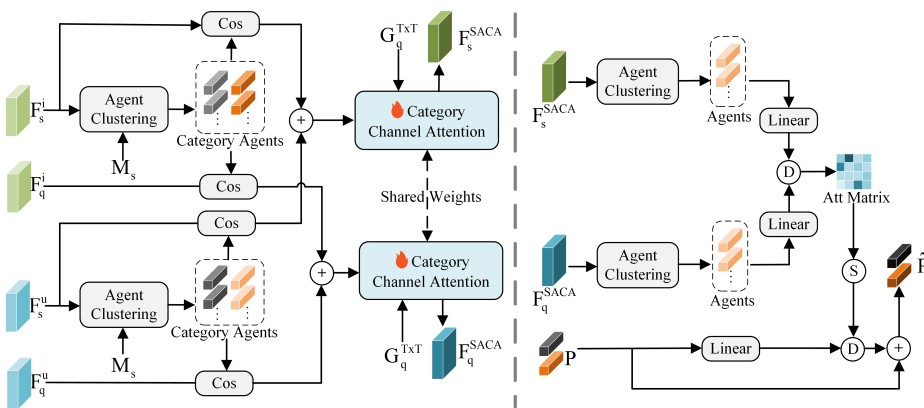

Figure 3: The architecture of the M-SACA and SAPA models. **M-SACA (Left)**: $\mathbf{F}_s^{i/u}$ use masks $\mathbf{M}_s$ to generate representative category agents through agent clustering, calculates cosine similarity with $\mathbf{F}_s^{i/u}$ and $\mathbf{F}_q^{i/u}$ respectively, and then generates intra-class correlation features through category channel attention. **SAPA (Right)**: $\mathbf{F}_{s/q}^{SACA}$ generate representative agents through agent clustering, form a cross-attention matrix, and then adaptively adjust the prototype through the attention matrix.

Next, calculate the cosine similarity between the query/support and the agent features to form the initial query/support correlation features:

$$\begin{cases} \mathbf{C}_q^i = \dfrac{\mathbf{F}_q^i \cdot \mathbf{A}_{agent}^{i\top}}{\left\|\mathbf{F}_q^i\right\|\left\|\mathbf{A}_{agent}^{i\top}\right\|}, & \mathbf{C}_q^u = \dfrac{\mathbf{F}_q^u \cdot \mathbf{A}_{agent}^{u\top}}{\left\|\mathbf{F}_q^u\right\|\left\|\mathbf{A}_{agent}^{u\top}\right\|}, \\[3mm] \mathbf{C}_s^i = \dfrac{\mathbf{F}_s^i \cdot \mathbf{A}_{agent}^{i\top}}{\left\|\mathbf{F}_s^i\right\|\left\|\mathbf{A}_{agent}^{i\top}\right\|}, & \mathbf{C}_s^u = \dfrac{\mathbf{F}_s^u \cdot \mathbf{A}_{agent}^{u\top}}{\left\|\mathbf{F}_s^u\right\|\left\|\mathbf{A}_{agent}^{u\top}\right\|}, \end{cases} \tag{3}$$

where $\mathbf{C}_q^i, \mathbf{C}_q^u \in \mathbb{R}^{N_q \times N_C \times N_A}$. $\mathbf{C}_s^i, \mathbf{C}_s^u \in \mathbb{R}^{N_s \times N_C \times N_A}$. Agent points can represent category information in the support set, enabling the construction of correlation features between support/query and agents features.

Use two linear layers to convert IF and UF correlations and merge them into multimodal correlation features:

$$\begin{cases} \hat{\mathbf{C}}_q = \mathcal{F}_{\text{lin1}}(\mathbf{C}_q^i) + \mathcal{F}_{\text{lin2}}(\mathbf{C}_q^u), \\ \hat{\mathbf{C}}_s = \mathcal{F}_{\text{lin1}}(\mathbf{C}_s^i) + \mathcal{F}_{\text{lin2}}(\mathbf{C}_s^u), \end{cases} \tag{4}$$

where $\hat{\mathbf{C}}_q$ and $\hat{\mathbf{C}}_s$ are combined by simply adding the transformed feature representations from different modalities. By introducing text semantic guidance $\mathbf{G}^{\text{TxT}}$, it can learn intra-class correlation features, which are then used to fuse multimodal information from text and vision:

$$\begin{cases} \mathbf{F}_q^{SACA} = \mathcal{F}_{\text{att}}(\hat{\mathbf{C}}_q, \mathbf{G}_q^{\text{TxT}}), \\ \mathbf{F}_s^{SACA} = \mathcal{F}_{\text{att}}(\hat{\mathbf{C}}_s, \mathbf{G}_s^{\text{TxT}}), \end{cases} \tag{5}$$

where $\mathbf{G}_q^{\text{TxT}} = \mathbf{F}_q^i \cdot \mathbf{T}^\top$ and $\mathbf{G}_s^{\text{TxT}} = \mathbf{F}_s^i \cdot \mathbf{T}^\top$. $\mathcal{F}_{\text{att}}$ is intra-class correlation attention, which uses the MSF module An et al. (2025b).

From this, we obtain the multimodal query/support correlation features $\mathbf{F}_q^{SACA}$ and $\mathbf{F}_s^{SACA}$.

### 3.4 SEMANTIC AGENTS PROTOTYPES ADAPTATION (SAPA)

As shown in Figure 3 (Right), the category prototype can be obtained through mean average pooling (MAP) as follows:

$$\mathbf{P} = \text{MAP}(\mathbf{F}_s^{SACA}, \mathbf{M}_s), \tag{6}$$

where $\mathbf{P} \in \mathbb{R}^{N_C \times D}$ represents the prototype.

To generate significant agent points, unlike Formula 2, there is no mask here. Therefore, instead of distinguishing between foreground and background, the focus is placed on the overall scene

structure, allowing the prototype to be adaptively adjusted based on global context. The definition is as follows:

$$\begin{cases} \mathbf{A}_{\mathrm{q}} = \mathcal{F}_{\mathrm{agent}}(\mathbf{F}_{\mathrm{q}}^{\mathrm{SACA}}, \mathbf{X}_{\mathrm{q}}), \mathbf{A}_{\mathrm{q}} \in \mathbb{R}^{N_P \times D}, \\ \mathbf{A}_{\mathrm{s}} = \mathcal{F}_{\mathrm{agent}}(\mathbf{F}_{\mathrm{s}}^{\mathrm{SACA}}, \mathbf{X}_{\mathrm{s}}), \mathbf{A}_{\mathrm{s}} \in \mathbb{R}^{N_P \times D}, \end{cases} \quad (7)$$

where $N_P$ represents the number of agents.

To calculate the channel cross-attention between the query/support agent correlation features $\mathbf{A}_{\mathrm{q}}$ and $\mathbf{A}_{\mathrm{s}}$, $\mathbf{Q}$, $\mathbf{K}$, and $\mathbf{V}$ are calculated separately using the following formulas:

$$\mathbf{Q} = \mathbf{A}_{\mathrm{q}}^{\top} \cdot \mathbf{W}^{\mathcal{Q}}, \quad \mathbf{K} = \mathbf{A}_{\mathrm{s}}^{\top} \cdot \mathbf{W}^{\mathcal{K}}, \quad \mathbf{V} = \mathbf{P} \cdot \mathbf{W}^{\mathcal{V}}, \quad (8)$$

where $\mathbf{W}^{\mathcal{Q}}$, $\mathbf{W}^{\mathcal{K}} \in \mathbb{R}^{N_P \times D'}$, $\mathbf{W}^{\mathcal{V}} \in \mathbb{R}^{D \times D}$ are learnable fully connected parameters that project $\mathbf{A}_{\mathrm{q}}$, $\mathbf{A}_{\mathrm{s}}$, and $\mathbf{P}$ onto the latent feature space. $D'$ is the dimension of the latent space features. $\mathbf{Q}$, $\mathbf{K} \in \mathbb{R}^{D \times D'}$ and $\mathbf{V} \in \mathbb{R}^{N_C \times D}$.

Then, calculate the channel attention map $w \in \mathbb{R}^{D \times D}$ through matrix multiplication:

$$w_{i,j} = \frac{\exp(\beta_{i,j})}{\sum_{j=1}^{D} \exp(\beta_{i,j})}, \quad \beta_{i,j} = \frac{\mathbf{Q} \cdot \mathbf{K}^{\top}}{\sqrt{D}}, \quad (9)$$

Cross-attention $w$ establishes a channel correspondence between query and support agent correlation features. Then, matrix multiplication is performed on $w$ and $\mathbf{V}$ to correct the prototype to adapt to the channel distribution between agent correlation features. Finally, the agent semantic adaptation prototype is obtained through a residual connection:

$$\hat{\mathbf{P}} = \mathbf{P} + (\mathbf{V} \cdot w)\mathbf{W}^{\mathcal{P}}, \quad (10)$$

where $\mathbf{W}^{\mathcal{P}} \in \mathbb{R}^{D \times D}$ is an adaptive fully connected parameter that adaptively adjusts the prototypes after cross-attention. By focusing attention on the features formed by query and support agent correlation, feature channels that better align with the query samples can be obtained, which helps to achieve better segmentation results.

We use the cosine similarity between the support feature $\mathbf{F}_{\mathrm{s}}^{\mathrm{SACA}}$ and the adaptive prototype $\hat{\mathbf{P}}$ to obtain the probability:

$$p^{\mathrm{SA}} = \tau \cdot \frac{\mathbf{F}_{\mathrm{q}}^{\mathrm{SACA}} \cdot \hat{\mathbf{P}}}{\left\|\mathbf{F}_{\mathrm{q}}^{\mathrm{SACA}}\right\| \left\|\hat{\mathbf{P}}\right\|}, \quad (11)$$

where $\tau$ is the scaling factor.

At the same time, the probability of the text modality can be obtained through text semantic guidance $\mathbf{G}_{\mathrm{q}}^{\mathrm{TxT}}$:

$$p^{\mathrm{TxT}} = \mathrm{Softmax}(\mathbf{G}_{\mathrm{q}}^{\mathrm{TxT}}). \quad (12)$$

$p^{\mathrm{TxT}}$ can achieve segmentation in the absence of samples, but introducing it directly into the probability of the visual modality (by accumulation or averaging) may confuse. Softmax is a row-based softmax function.

Hence, we created a discriminative deconfusion mechanism that combines text and visual modalities. It mainly learns deconfusion from text predictions through a residual structure, and then fuses them by generating adaptive weights.

## 3.5 DISCRIMINATIVE DECONFUSION (DD)

$p^{\mathrm{TxT}}$ as prior knowledge may contain a lot of interfering information, affecting intra-class/inter-class recognition. Therefore, we use $\mathcal{E}_{\mathcal{A}2}$ to remove interference factors from $p^{\mathrm{TxT}}$. $\mathcal{E}_{\mathcal{A}1}$ primarily extracts visual–textual discriminative information from $\mathbf{F}_{\mathrm{q}}^{\mathrm{SACA}}$, then integrates it by $\mathcal{E}_{\mathcal{A}3}$ to reduce the discriminative confusion introduced by the residual connection:

$$p^{\mathrm{DD}} = p^{\mathrm{TxT}} + \mathcal{E}_{\mathcal{A}3}\left(\mathcal{E}_{\mathcal{A}2}(p^{\mathrm{TxT}}) + \mathcal{E}_{\mathcal{A}1}(\mathbf{F}_{\mathrm{q}}^{\mathrm{SACA}})\right). \quad (13)$$

Finally, the text prediction, which mitigates inter-class confusion, is fused with the visual prediction using an adaptive weighting strategy. The adaptive weights are produced by the generator $\mathcal{G}$, which takes $\mathbf{F}_q^{\text{SACA}}$ as input. Specifically, $\mathcal{G}$ is a lightweight multilayer perceptron composed of a Linear layer, followed by LayerNorm, a ReLU activation, Dropout, another Linear layer, and a final Sigmoid function:

$$p^{\text{F}} = \alpha p^{\text{SA}} + (1 - \alpha)p^{\text{DD}}, \quad \alpha = \mathcal{E}_{\mathcal{G}}(\mathbf{F}_q^{\text{SACA}}), \tag{14}$$

where $\alpha \in \mathbb{R}^{N_q \times 1}$ generates dynamic weights for each query point in the query set.

## 3.6 Training Objective

During the training phase, cross-entropy (CE) and L1 similarity loss are used together for optimization. Except for text prediction, other predictions are learned under the supervision of the CE function:

$$\mathcal{L}_{CE} = \mathcal{L}_{CE}^{\text{SA}} + \mathcal{L}_{CE}^{\text{DD}} + \mathcal{L}_{CE}^{\text{F}}, \tag{15}$$

where $\mathcal{L}_{CE}^{\text{SA}} = \mathcal{L}_{CE}(p^{\text{SA}}, \mathbf{M}_q), \mathcal{L}_{CE}^{\text{DD}} = \mathcal{L}_{CE}(p^{\text{DD}}, \mathbf{M}_q), \mathcal{L}_{CE}^{\text{F}} = \mathcal{L}_{CE}(p^{\text{F}}, \mathbf{M}_q)$.

To align text and visual modalities, L1 similarity is used to constrain visual predictions:

$$\mathcal{L}_{\text{sim}} = \mathcal{L}_{\text{sim}}^{\text{DD}} + \mathcal{L}_{\text{sim}}^{\text{SA}}, \tag{16}$$

where $\mathcal{L}_{\text{sim}}^{\text{DD}} = \left\| p^{\text{TxT}} - p^{\text{DD}} \right\|_1, \mathcal{L}_{\text{sim}}^{\text{SA}} = \left\| p^{\text{TxT}} - p^{\text{SA}} \right\|_1$.

The final loss function is defined as follows:

$$\mathcal{L} = \mathcal{L}_{CE} + \mathcal{L}_{\text{sim}}. \tag{17}$$

## 4 Experiments

### 4.1 Implementation Details

The backbone we use comes from the first two layers of the Stratified Transformer Lai et al. (2022), and both the UF Head and IF Head are consistent with the third layer structure. To eliminate bias in the base category and improve generalization ability, we also used TACC An et al. (2025b) during the testing phase. $\tau$ is set to 10 by default He et al. (2023). According to COSeg, in the 2-way setting, each category has 100 episodes, and in the 1-way setting, each category has 1000 episodes. The pre-training stage is conducted for 100 epochs, followed by 60 epochs in the meta-learning Ren et al. (2018) stage. The AdamW optimizer is employed with a weight decay of 0.01. The learning rates are set to 0.006 for pre-training and 0.0001 for meta-learning. The number of agents in M-SACA is 100 for each category, and the number of agent points in SAPA is 512.

### 4.2 Datasets

Use the two most popular datasets for FS-PCS tasks, S3DIS Armeni et al. (2016), ScanNet Dai et al. (2017), and SemanticKITTI Behley et al. (2019). Slice the scene point cloud along the $XY$ axis into 1m×1m blocks with a stride of 1m. Voxelize each block with a grid size of 0.02m, using sampling to maintain a maximum of 20,480 points per block. In visualization, we set the scene point cloud to be sliced into 2m×2m blocks along the $XY$ axis, with a stride of 2m. We also used 0.02m grid voxelization, keeping each block to a maximum of 81,920 points.

### 4.3 Comparison with the State-of-the-art

We compared MAD with previous models on the S3DIS, ScanNet, and SemanticKITTI datasets, with detailed comparisons shown in Tables 1, 2, 4b, and 4c. The comparison methods include attMPTI Zhao et al. (2021b), QGE Ning et al. (2023), QGPA He et al. (2023), COSeg An et al. (2024), COSeg[†] An et al. (2024), and MM-FSS An et al. (2025b). COSeg[†], MM-FSS, and MAD all use the same 2D-aligned pre-trained backbone.

Overall, MAD significantly outperforms (by >3%) the previous state-of-the-art technique (MM-FSS) across all settings in both datasets. These results demonstrate that MAD is a more effective multimodal data fusion method, significantly improving the segmentation of novel categories.

| Methods | 1-way 1-shot | | | 1-way 5-shot | | | 2-way 1-shot | | | 2-way 5-shot | | |
|---|---|---|---|---|---|---|---|---|---|---|---|---|
| | $S^0$ | $S^1$ | Mean | $S^0$ | $S^1$ | Mean | $S^0$ | $S^1$ | Mean | $S^0$ | $S^1$ | Mean |
| AttMPTI (CVPR 2021) | 36.32 | 38.36 | 37.34 | 46.71 | 42.70 | 44.71 | 31.09 | 29.62 | 30.36 | 39.53 | 32.62 | 36.08 |
| QGE (ACMM 2023) | 41.69 | 39.09 | 40.39 | 50.59 | 46.41 | 48.50 | 33.45 | 30.95 | 32.20 | 40.53 | 36.13 | 38.33 |
| QGPA (TIP 2023) | 35.50 | 35.83 | 35.67 | 38.07 | 39.70 | 38.89 | 25.52 | 26.26 | 25.89 | 30.22 | 32.41 | 31.32 |
| COSeg (CVPR 2024) | 46.31 | 48.10 | 47.21 | 51.40 | 48.68 | 50.04 | 37.44 | 36.45 | 36.95 | 42.27 | 38.45 | 40.36 |
| COSeg† (CVPR 2024) | 47.17 | 48.37 | 47.77 | 50.93 | 49.88 | 50.41 | 37.15 | 38.99 | 38.07 | 42.73 | 40.25 | 41.49 |
| MM-FSS (ICLR 2025) | 49.84 | 54.33 | 52.09 | 51.95 | 56.46 | 54.21 | 41.98 | 46.61 | 44.30 | 46.02 | 54.29 | 50.16 |
| MAD (ours) | 52.10±0.35 | 58.27±0.27 | 55.19(+3.1) | 58.63±0.17 | 62.13±0.13 | 60.38(+6.2) | 50.13±0.37 | 52.23±0.29 | 50.77(+6.5) | 49.05±0.29 | 58.59±0.32 | 53.82(+3.7) |

Table 1: Comparison of few-shot quantitative results based on the S3DIS dataset. The best results are highlighted in bold. $S^0$ and $S^1$ denote two different splits. † represents the introduction of multimodal data.

| Methods | 1-way 1-shot | | | 1-way 5-shot | | | 2-way 1-shot | | | 2-way 5-shot | | |
|---|---|---|---|---|---|---|---|---|---|---|---|---|
| | $S^0$ | $S^1$ | Mean | $S^0$ | $S^1$ | Mean | $S^0$ | $S^1$ | Mean | $S^0$ | $S^1$ | Mean |
| AttMPTI (CVPR 2021) | 34.03 | 30.97 | 32.50 | 39.09 | 37.15 | 38.12 | 25.99 | 23.88 | 24.94 | 30.41 | 27.35 | 28.88 |
| QGE (ACMM 2023) | 37.38 | 33.02 | 35.20 | 45.08 | 41.89 | 43.49 | 26.85 | 25.17 | 26.01 | 28.35 | 31.49 | 29.92 |
| QGPA (TIP 2023) | 34.57 | 33.37 | 33.97 | 41.22 | 38.65 | 39.94 | 21.86 | 21.47 | 21.67 | 28.02 | 29.34 | 29.18 |
| COSeg (CVPR 2024) | 41.73 | 41.82 | 41.78 | 48.31 | 44.11 | 46.21 | 28.72 | 28.78 | 28.75 | 35.97 | 33.39 | 34.68 |
| COSeg† (CVPR 2024) | 41.95 | 42.07 | 42.01 | 48.54 | 44.68 | 46.61 | 29.54 | 28.51 | 29.03 | 36.87 | 34.15 | 35.51 |
| MM-FSS (ICLR 2025) | 46.08 | 43.37 | 44.73 | 54.66 | 45.48 | 50.07 | 43.99 | 34.43 | 39.21 | 48.86 | 39.32 | 44.09 |
| MAD (ours) | 52.19±0.25 | 45.67±0.27 | 48.93(+4.2) | 59.41±0.10 | 48.39±0.32 | 53.90(+3.8) | 49.15±0.24 | 40.50±0.13 | 44.83(+5.6) | 49.86±0.14 | 53.10±0.28 | 51.48(+7.4) |

Table 2: Comparison of few-shot quantitative results based on the ScanNet dataset. The best results are highlighted in bold.

Specifically, on the S3DIS dataset, MAD improved by 4.7% and 5.1% in the 1-way and 5-way settings, respectively. On the ScanNet dataset, improvements of 4% and 6.5% were achieved in the 1-way and 5-way settings, respectively. Figure 4 shows that MM-FSS has large areas of misclassification (identifying the background as the foreground category) in the segmentation of the new category. MAD consistently achieves accurate segmentation, effectively mitigating discriminative confusion and demonstrating strong capability in segmenting novel categories.

| DD | M-SACA | SAPA | mIoU |
|---|---|---|---|
| | | | 41.80 |
| ✓ | | | 43.06 |
| ✓ | ✓ | | 43.46 |
| | ✓ | ✓ | 48.08 |
| ✓ | ✓ | ✓ | **50.13** |

(a)

| $\mathcal{E}_{\mathcal{A}1}$ | $\mathcal{E}_{\mathcal{A}2}$ | $\mathcal{E}_{\mathcal{A}3}$ | Res | mIoU |
|---|---|---|---|---|
| ✓ | ✓ | | ✓ | 49.65 |
| ✓ | | ✓ | ✓ | 49.79 |
| | ✓ | ✓ | ✓ | 48.72 |
| ✓ | ✓ | ✓ | | 47.68 |
| ✓ | ✓ | ✓ | ✓ | **50.13** |

(b)

| $\mathcal{L}_{CE}^{F}$ | $\mathcal{L}_{CE}^{DD}$ | $\mathcal{L}_{CE}^{SA}$ | mIoU |
|---|---|---|---|
| ✓ | | | 47.42 |
| ✓ | ✓ | | 47.53 |
| ✓ | | ✓ | 48.05 |
| ✓ | ✓ | ✓ | **48.13** |

(c)

| UF | IF | Text | mIoU |
|---|---|---|---|
| ✓ | | | 40.18 |
| ✓ | ✓ | | 45.30 |
| ✓ | | ✓ | 49.31 |
| ✓ | ✓ | ✓ | **50.13** |

(d)

| logits | mIoU |
|---|---|
| Average | 47.17 |
| Sum | 44.23 |
| Generator | **50.13** |

(e)

| $\mathcal{L}_{sim}^{SA}$ | $\mathcal{L}_{sim}^{DD}$ | mIoU |
|---|---|---|
| ✓ | | 46.04 |
| | ✓ | 46.00 |
| ✓ | ✓ | **50.13** |

(f)

| Methods | FLOPs | Params |
|---|---|---|
| MM-FSS | **5.54G** | **9.90M** |
| MAD (ours) | 8.44G | 10.54M |

(g)

Table 3: Ablation Study(%). (a) Impact of the Sub-module. (b) Ablation on the DD. (c) (f) Impact of the Training Objective. (d) Contribution of Each Modality. (e) Influence of the Generator in DD. (g) Model Complexity Analysis.

## 4.4 ABLATION STUDY

In this section, unless otherwise specified, all results are mIoU results on the S3DIS dataset using 2-way 1-shot.

**Impact of the Sub-module.** Table 3a shows the results of different submodules. DD, M-SACA, and SAPA can all improve mIoU to a certain extent. Among them, SAPA can significantly (+6.67%) improve the segmentation effect. M-SACA is the pre-module of SAPA, enabling segmentation us-

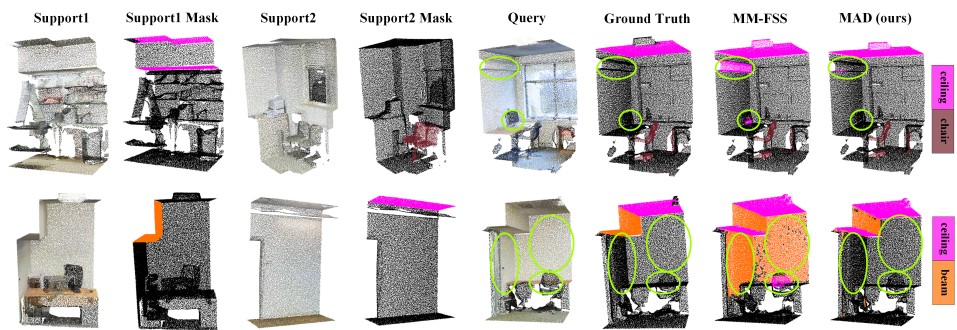

Figure 4: Qualitative results of MM-FSS and MAD methods based on the S3DIS dataset visualized in the 2-way 1-shot setting.

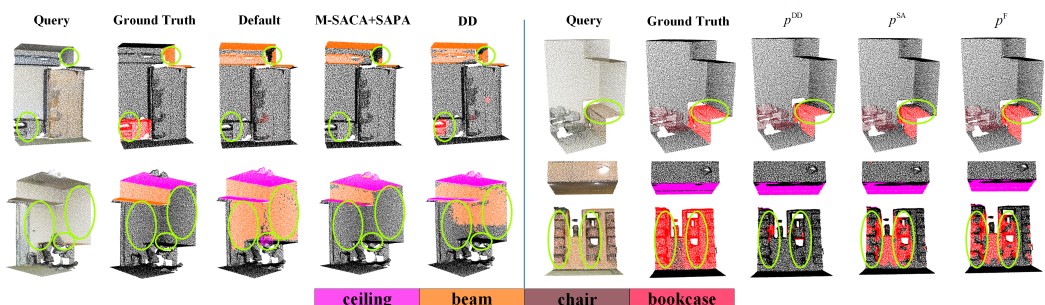

Figure 5: Visualization results based on the S3DIS dataset in the 2-way 1-shot setting. The left side shows the results of adding M-SACA+SAPA or DD separately. On the right are the predicted results for $p^{DD}$, $p^{SA}$, and $p^{F}$.

ing correlation features directly in multimodal settings. **Ablation on the DD.** Table 3b provides the results of internal ablation studies of the DD module. The use of adapters $\mathcal{E}_A$ can improve performance to a certain extent, with the residual structure making a significant contribution (+2.45%). **Impact of the Training Objective.** Tables 3c and 3f present the results of different training objective combinations. Using $\mathcal{L}_{CE}^{F}$ alone already yields a good mIoU. Both $\mathcal{L}_{CE}^{DD}$ and $\mathcal{L}_{CE}^{SA}$ lead to moderate improvements. It is worth noting that removing $\mathcal{L}_{sim}^{SA}$ or $\mathcal{L}_{sim}^{DD}$ results in a sharp drop in mIoU by approximately 4%. **Contribution of Each Modality.** Table 3d shows the results of combining different modalities. Compared to using only UF Head data, the introduction of IF Head and text data leads to significant improvements (+5.12% / +9.13%). These results demonstrate that MAD effectively integrates image and text modalities with point cloud data, highlighting its potential for segmenting novel categories. **Influence of the Generator in DD.** Table 3e provides the ablation results of the generator on DD. Compared with averaging or summing, the generator improved by 2.96% and 5.9%, respectively. **Model Complexity Analysis.** Table 3g presents a detailed comparison between our method and the state-of-the-art MM-FSS approach. Our method achieves a substantial improvement in mIoU with only a minor increase in parameters and computational complexity. **Impact of M-SACA+SAPA and DD.** As shown in the Figure 5, compared to default, M-SACA+SAPA and DD both mitigate the misclassification of background points as foreground points, and M-SACA+SAPA can achieve correct classification even in scenarios where DD also misclassifies. Only DD significantly improves the identification of challenging foreground categories. Compared with $p^{DD}$ and $p^{SA}$, $p^{F}$ can effectively expand and maintain intra-class consistency (right side). **More Visualizations.** Figure 6 (left) demonstrates the principle and effectiveness of the $\mathcal{G}$. In contrast to averaging or summation, the adaptive weighting of the generator integrates prediction results more effectively, leading to more accurate segmentation. Figure 6 (right) shows that $p^{TxT}$ provides only a rough approximation of semantics, containing numerous prior errors. In contrast, $p^{DD}$ effectively corrects these errors, yielding superior results. **Impact of CLIP Deficiency.** Table 4a demonstrates that MAD outperforms existing state-of-the-art methods even without incorporating CLIP priors, validating MAD's superiority. **Analysis of Modal Gap.** Figure 7 shows a significant

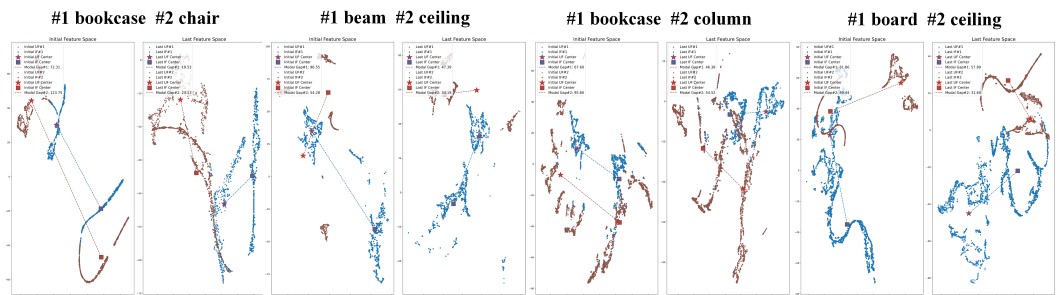

Figure 6: Visualization results based on the S3DIS dataset in the 2-way 1-shot setting. The left side shows the predictions for $p^{TxT}$ and $p^{DD}$. On the right is the visualization for the generator, average, and sum.

Figure 7: Visualization results of the Modal Gap under the S3DIS dataset in the 2-way 1-shot setting. We use t-SNE to project high-dimensional features onto a 2D display, visualizing the cluster centers for each category and the gaps between them.

modal gap in the initial stage. Our method can substantially reduce this gap in the last stage, thereby achieving effective alignment of data across different modalities.

(a)

| Model | CLIP | mIoU |
|---|---|---|
| COSeg | | 37.44 |
| COSeg† | ✓ | 37.15 |
| MM-FSS | ✓ | 41.98 |
| MAD (ours) | | **43.59** |
| MAD (ours) | ✓ | **50.13** |

(b)

| Methods | 1-way 1-shot | | |
|---|---|---|---|
| | $S^0$ | $S^1$ | Mean |
| MM-FSS | 31.73 | 30.79 | 31.26 |
| MAD (ours) | 33.70 | 38.72 | **36.21** |

(c)

| Methods | 2-way 1-shot | | |
|---|---|---|---|
| | $S^0$ | $S^1$ | Mean |
| MM-FSS | 21.30 | 30.38 | 25.84 |
| MAD (ours) | 25.99 | 36.23 | **31.16** |

Table 4: (a) Impact of CLIP Deficiency. (b) (c) Quantitative results on the SemanticKITTI.

## 5 CONCLUSION

In this paper, we propose a novel method named MAD that can effectively fuse multimodal information to improve FS-PCS. MAD mainly consists of M-SACA, SAPA, and DD modules. M-SACA generates correlation features through mask agent points, integrates visual and textual information, and rebuilds the correlation features of the query and support set. SAPA enhances the adaptive representation capability of prototypes by correcting them through maskless agent points. DD effectively alleviates the discriminatory confusion problem and eliminates a priori bias. The proposed method significantly improves segmentation performance on the FS-PCS dataset, validating its effectiveness in addressing FS-PCS issues.

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
