## A   MORE EXPERIMENTS

More quantitative results based on the SemanticKITTI Behley et al. (2019) dataset are shown in Table 5, demonstrating that our method remains the best across all settings. Furthermore, compared to the indoor dataset, increasing the number of shots from 1 to 5 in the same way does not significantly improve segmentation performance (compare Tables 4b, 4c, and 5). Especially in the 2-way test, comparing Tables 4c and 4c, the average score improved from 31.16% to 31.53%. This marginal gain of 0.37% underscores the challenges posed by SemanticKITTI, which arise from its sparse point distribution and lack of color information. Further research on such data is required to enable practical applications.

| Methods | 1-way 5-shot | | |
| --- | --- | --- | --- |
| | $S^0$ | $S^1$ | Mean |
| MM-FSS | 34.61 | 40.83 | 37.72 |
| MAD (ours) | 35.72 | 42.23 | **38.98** |

(a)

| Methods | 2-way 5-shot | | |
| --- | --- | --- | --- |
| | $S^0$ | $S^1$ | Mean |
| MM-FSS | 25.69 | 30.19 | 27.94 |
| MAD (ours) | 28.72 | 34.34 | **31.53** |

(b)

Table 5: (a) (b) More Quantitative results on the SemanticKITTI.

## B   MORE DETAILS

### B.1   PRE-TRAINING DETAILS

The pre-training phase primarily trains the IF Head to extract 3D features aligned with the image. The trained image encoders (LSeg Li et al. (2022) and CLIP Radford et al. (2021)) extract 2D features $f^{2d} \in \mathbb{R}^{H \times W \times D_t}$, and the backbone and IF Head are used to extract 3D features $f^{3d} \in \mathbb{R}^{N \times D_t}$. To enable the IF Head to represent image features, we use $f^{2d}$ to supervise $f^{3d}$. Therefore, it is necessary to establish the correspondence between pixels and points, that is, to match the point $p \in \mathbb{R}^3$ with the pixel $\mathbf{x} = (x, y)$ in the image. In MM-FSS, the 2D pixel space is projected to the 3D space via $\mathbf{x} = M_{int} \times M_{ext} \times p$, and optimized using a cosine similarity loss (Peng et al. (2023)). Here, $M_{int}$ and $M_{ext}$ are the internal and external matrices, respectively, from camera-to-pixel and world-to-camera.

Only the ScanNet Dai et al. (2017) dataset provides 2D images and camera parameters; the RGB images typically have a resolution of $1920 \cdot 1080$. Without any manually labeled data, we pre-train a skeleton network and an IF head using images and point cloud data from the ScanNet dataset. Subsequently, pre-trained weights are used directly for meta-learning. Note that during the subsequent formal training phase, the backbone network and IF Head are frozen to preserve the image representation.

### B.2   IMPLEMENTATION DETAILS OF $\mathcal{F}_{\text{AGENT}}$

Our $\mathcal{F}_{\text{agent}}$ is implemented using a non-parametric feature clustering method seeded by sampling the farthest point, eliminating the need for manual adjustment of the cluster radius. Given point coordinates $\{\mathbf{x}_i \in \mathbb{R}^3\}_{i=1}^N$, and features, taking the generation of $N_p$ prototypes as an example, the specific algorithm flow is as follows:

#### B.2.1   FARTHEST POINT SAMPLING

First, farthest point sampling is performed based on the 3D coordinates $\mathbf{x}_i$, obtaining $N_p$ seed indices $s_j \in \{1, \ldots, N\}$. These seeds preserve the main structure of the 3D scene. $N$ is the number of points.

#### B.2.2   NEAREST NEIGHBOR SEED POINT ALLOCATION

Then, adjacency and clustering are determined solely by nearest-prototype relationships in the feature space, rather than by a fixed radius. Specifically, we compute the distance between each point

feature and each seed feature:

$$d_{i,j} = \left\| \mathbf{f}_i - \mathbf{f}_{s_j} \right\|_2, \quad i = 1, \ldots, N, \ j = 1, \ldots, N_p. \tag{18}$$

$\| \cdot \|_2$ represents calculating the Euclidean distance, and $d_{i,j}$ represents the distance from point $i$ to seed $j$. Which can assign each point to its nearest seed point:

$$a_i = \arg \min_{1 \le j \le N_p} d_{i,j}, \tag{19}$$

$a_i$ represents the seed point number to which point $i$ belongs. This can form $N_p$ clusters $\{\mathcal{C}_j\}_{j=1}^{N_p}$ corresponding to different seed points. For example, the cluster associated with the $j$-th seed is as follows:

$$\mathcal{C}_j = \{i \mid a_i = j\}. \tag{20}$$

$\mathcal{C}_j$ represents the set of points belonging to seed $j$. Therefore, adjacency belongs to the same nearest seed cluster, which is learned based on the feature metric and has no additional radius hyperparameter.

### B.2.3 PROTOTYPE AGGREGATION AND REVERSE ASSIGNMENT

Generally, for each cluster $\mathcal{C}_j$, the prototype is calculated using the average. If a seed point is not assigned to any point (indicating an outlier), we fall back to using the seed feature itself as the prototype, as defined below:

$$\mathbf{p}_j = \begin{cases} \dfrac{1}{|\mathcal{C}_j|} \sum_{i \in \mathcal{C}_j} \mathbf{f}_i, & \text{if } |\mathcal{C}_j| > 0, \\ \mathbf{f}_{s_j}, & \text{if } |\mathcal{C}_j| = 0. \end{cases} \tag{21}$$

$\mathbf{p}_j$ represents the cluster prototype $j$, and $| \cdot |$ represents the number of elements in the set.

### B.2.4 BOUNDARY CASES AND MISSING VALUES

Invalid points or NaN values are filtered out during preprocessing and therefore will not enter the prototype module. When the number of feature points is less than the required number of prototypes, we explicitly pad the feature set with zero vectors to ensure that the module always returns $N_p$ prototypes. This design aims to avoid numerical instability. Since the clustering step relies solely on distances and assumes no specific data distribution, the method is robust to variations in the underlying distribution.

### B.3 TIME AND MEMORY CONSUMPTION

To further illustrate the overhead of our method, we report the time and memory consumption of MAD. Tested on the S3DIS Armeni et al. (2016) dataset (2-way 1-shot fold=0) and implemented on four RTX A6000 (48GB ) GPUs. As shown in Table 6, MAD consumes slightly more time and memory than the current state-of-the-art method (MM-FSS An et al. (2025b)). Unlike MM-FSS, the primary source of increased memory consumption is the learning of the supporting-set correlation features $F_s^{SACA}$. Further exploration of lightweight models is needed to facilitate deployment in real-world applications.

| Methods | Times (s) | GPU Memory (GB) |
|---|---|---|
| MM-FSS | 0.624 | 23.76 |
| MAD (ours) | 0.675 | 35.01 |

Table 6: Time and memory consumption. The average time per batch and the maximum memory usage during the first epoch.

### B.4 Hyperparameter Details

$D'$ denotes the dimension of the latent space, which is set equal to the input feature dimension $D$, following QGPA He et al. (2023). Consistent with previous work He et al. (2023); Ning et al. (2023), we set $\tau = 10$ by default.

### B.5 UF and IF Heads

To obtain a unified representation, a Unimodal Feature (UF) Head and an Intermodal Feature (IF) Head are incorporated into the 3D point cloud framework. The IF Head primarily integrates 2D visual information into the 3D point cloud by leveraging image encoding as supervision during pre-training An et al. (2025b). During the training phase, both the backbone and the IF Head are frozen to preserve feature space alignment between the 2D and 3D modalities.

### B.6 Generator and MLP Adapter Details

As shown in Figure 8, the adapter MLP undergoes a linear layer, followed by a layer normalization. A ReLU activation, and subsequently passes through dropout, before finally passing through a linear layer. The dropout rate is set to 0.1. In contrast, the output dimension of the last linear layer of the generator is 1, and it has undergone the sigmoid function.

## C Motivation

The incorporation of multimodal data has introduced significant potential to FS-PCS, highlighting the importance of effectively integrating information across different modalities. MM-FSS An et al. (2025b) employs the MSF module to learn and fuse category-level correlation features from multiple modalities. It then applies segmentation heads to perform point cloud segmentation, achieving strong performance. However, this limits the model's representational capabilities to a certain extent, so we considered how to perform segmentation directly using correlation features. Our experiments reveal a significant issue of discriminative confusion in multimodal FS-PCS tasks, where both intra-class and inter-class ambiguities substantially impact accuracy. How to discriminate confusion in such settings is what we expect to solve.

## D Additional Visualize Results

Figures 9 and 10 show a visual comparison between our method and the state-of-the-art methods. The main issues with MM-FSS include: By mistakenly dividing the background into foreground categories (Figure 2), MAD can alleviate these problems. In terms of complete segmentation of foreground objects, MAD shows significant improvement over MM-FSS. As shown in the last row of Figure 10, MM-FSS fails to recognize the bookcase, while MAD is able to recognize it correctly.

As shown in Figures 11 and 12, the M-SACA+SAPA module and DD module have significantly improved semantic category recognition compared to the default (as the second and third rows). The default incorrectly identifies the bookcase as the ceiling, while the M-SACA+SAPA and DD modules can identify the correct category. However, there is room for improvement in terms of object integrity.

As shown in Figure 13, the prediction $p^F$ can effectively integrate $p^{SA}$ and $p^{DD}$ predictions (first row), but as shown in the second row, it demonstrates inappropriate integration, achieving both correct and incorrect expansions when implementing expansion. This is also a focus of subsequent research. Further visualizations of the generator, $p^{DD}$, and modal gap are shown in Figures 14, 15, and 16.

## E Additional Discussions

Although our approach achieves state-of-the-art performance on FS-PCS under stringent evaluation protocols An et al. (2024), substantial variation in scene characteristics necessitates adaptation to

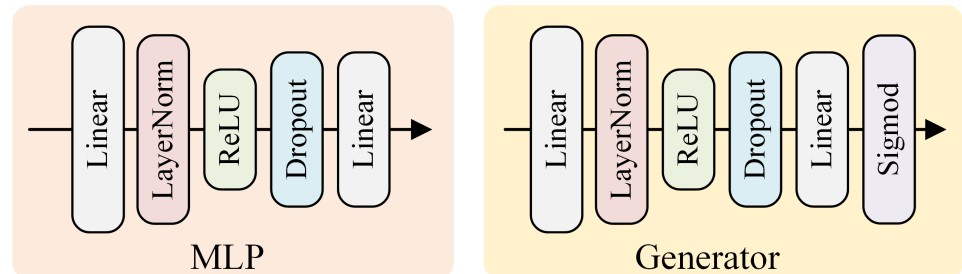

Figure 8: Generator and MLP Adapter Details.

specific deployment scenarios. This challenge is hindered by the scarcity of large-scale 3D point cloud datasets, which are far less abundant than image and video data, and may require the broader adoption of LiDAR acquisition systems. Moreover, despite maintaining reasonable computational efficiency, practical applications demand lightweight models to meet real-time latency constraints. Addressing these constraints will be a key direction for future work to enable the practical deployment of FS-PCS.

## F CONTRIBUTION STATEMENT

In general, existing methods Zhao et al. (2021b); Ning et al. (2023); He et al. (2023); Wang et al. (2025), are either unimodal and not applicable to real-world scenarios An et al. (2024), or they restrict multimodal data representation An et al. (2025b). Unlike these methods, to our knowledge, our MAD method is the first study to explore the simultaneous fusion of query and support point cloud multimodal correlation features in FS-PCS tasks. This will promote research in the community.

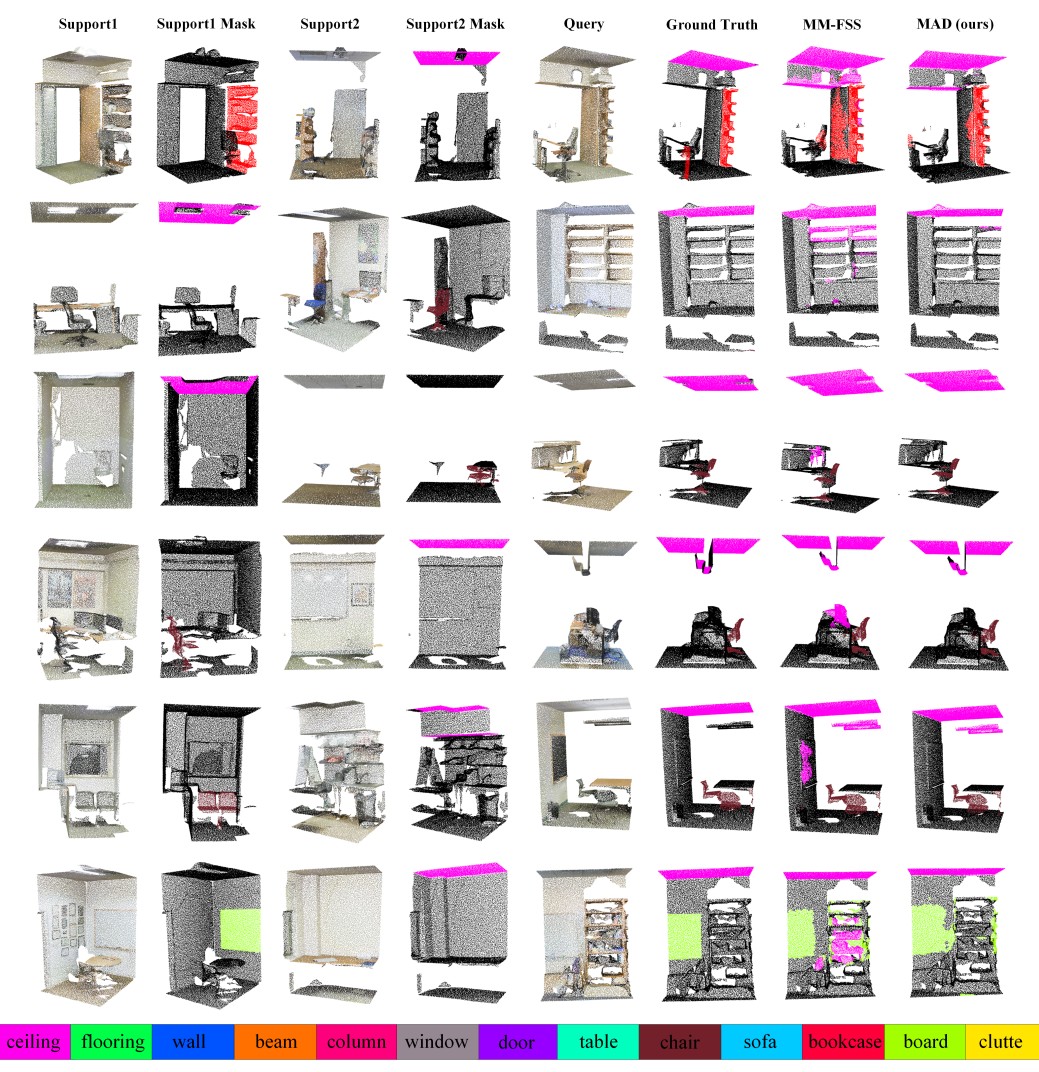

Figure 9: Visualization comparison results with MM-FSS on the S3DIS dataset. Default setting is 2-way 1-shot.

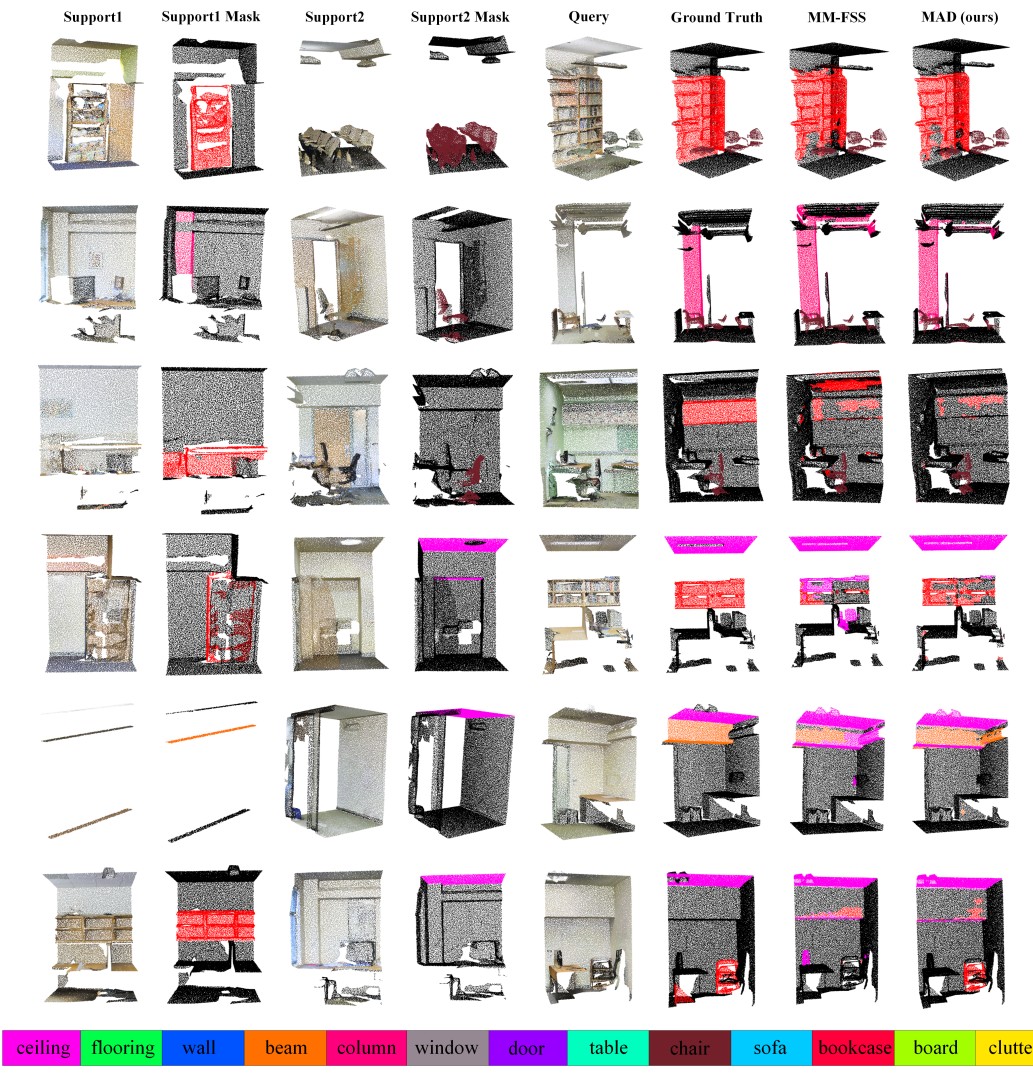

Figure 10: More visualization comparison results with MM-FSS on the S3DIS dataset. Default setting is 2-way 1-shot.

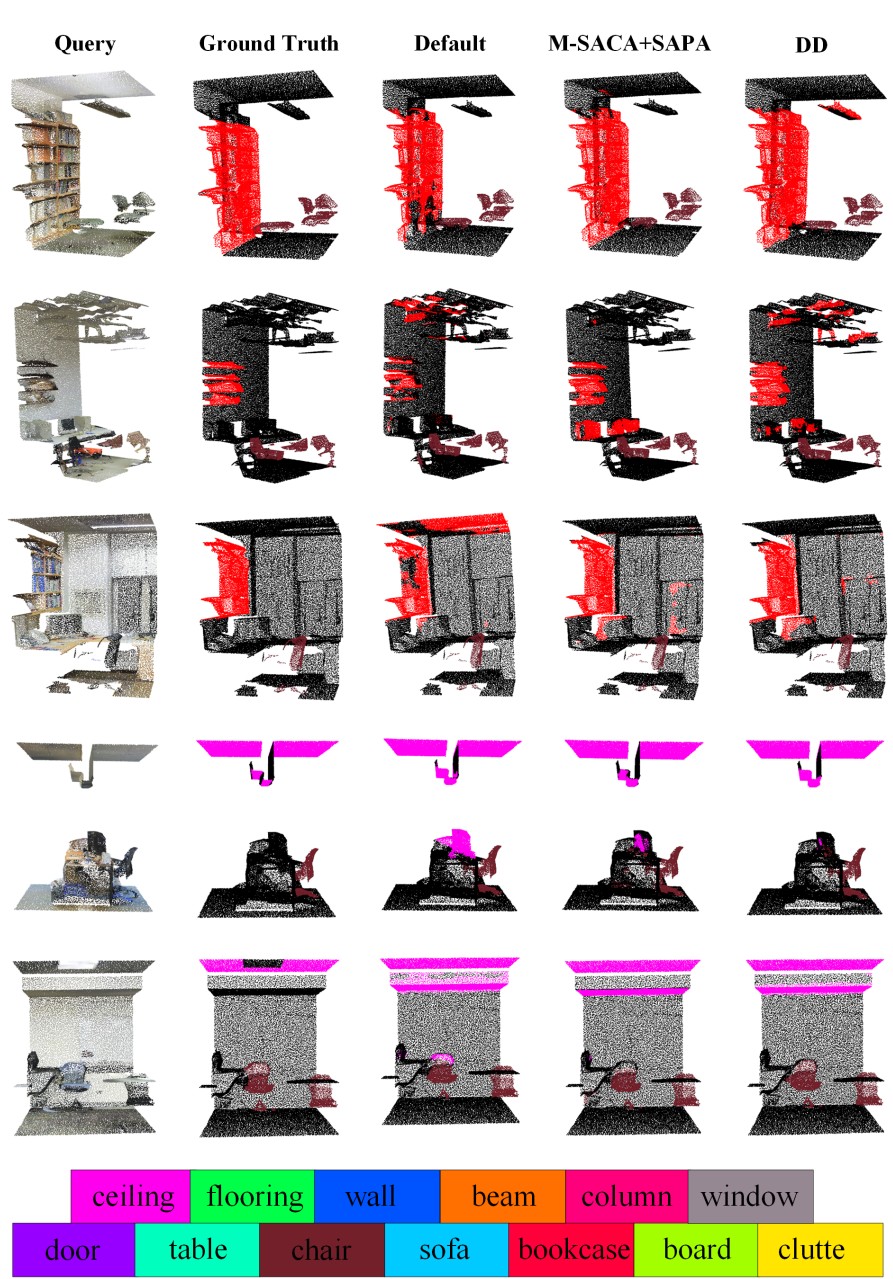

Figure 11: Visualization results based on the S3DIS dataset in the 2-way 1-shot setting. The results of adding M-SACA+SAPA or DD separately.

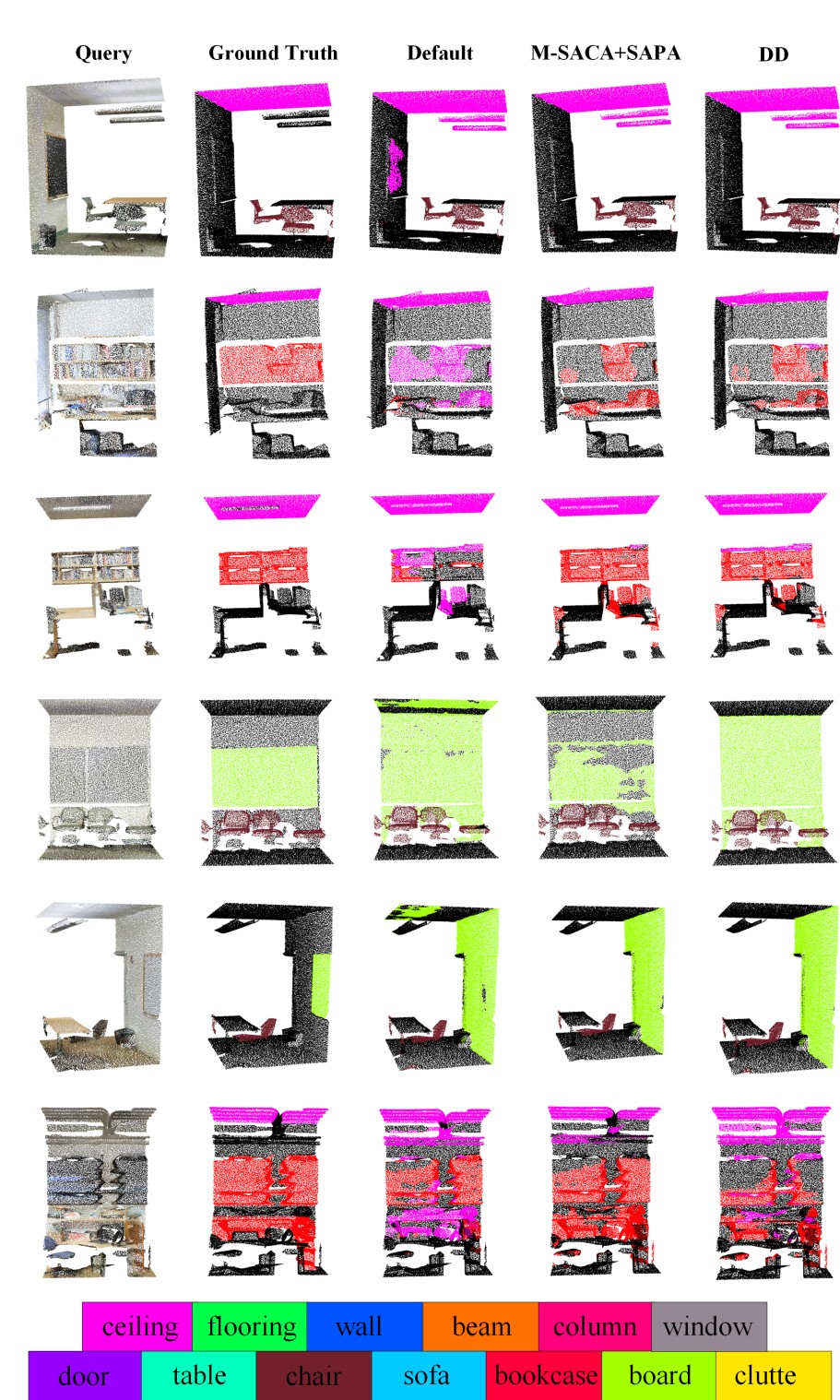

Figure 12: Visualization results based on the S3DIS dataset in the 2-way 1-shot setting. The results of adding M-SACA+SAPA or DD separately.

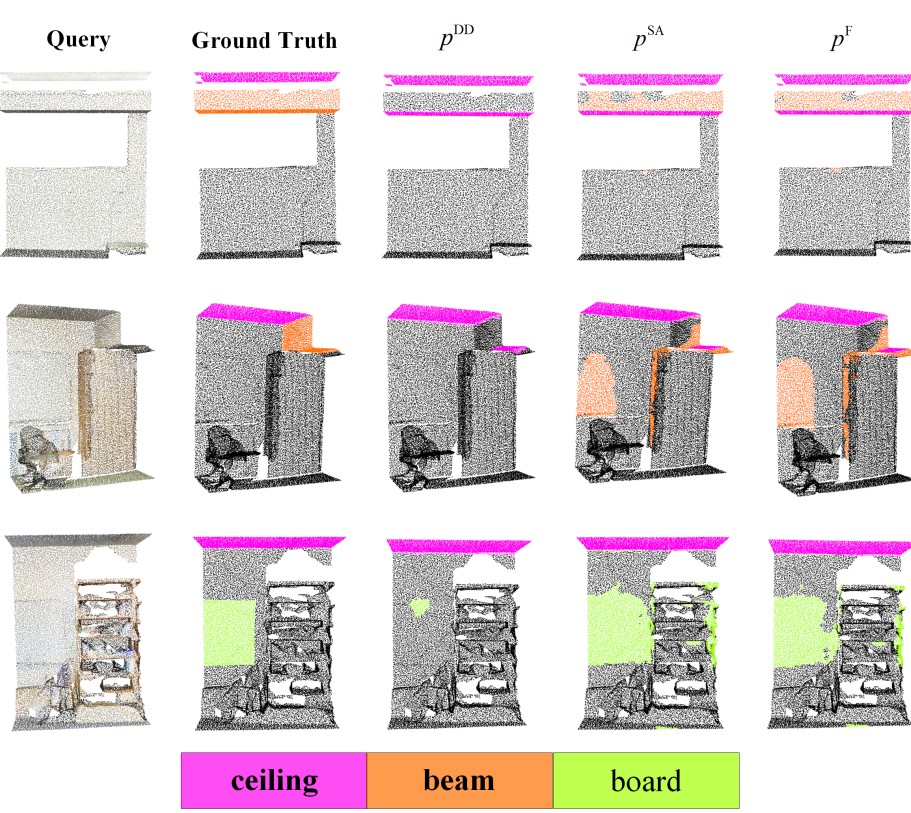

Figure 13: Visualization results based on the S3DIS dataset in the 2-way 1-shot setting. The predicted results for $p^{\text{DD}}$, $p^{\text{SA}}$, and $p^{\text{F}}$.

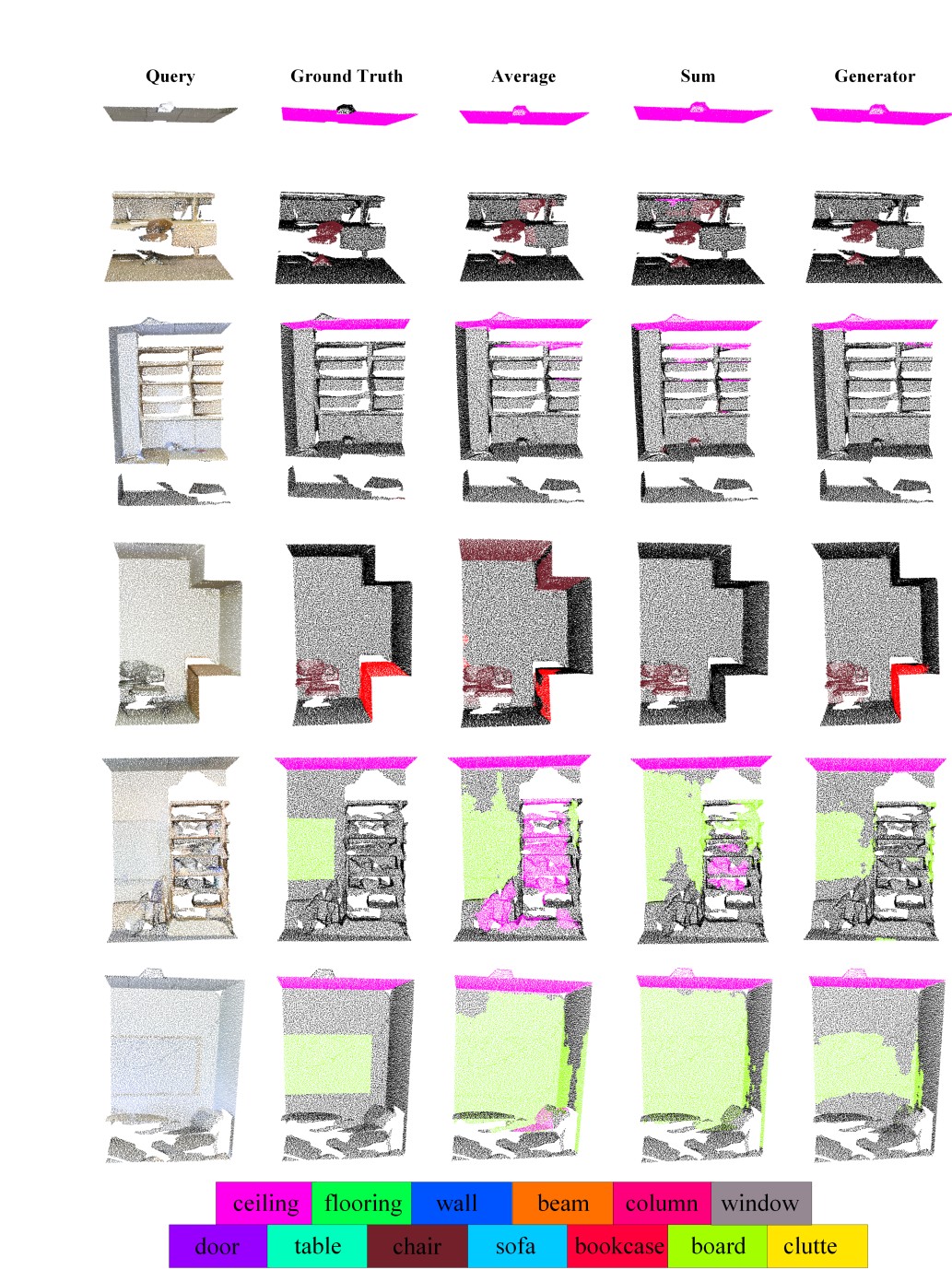

Figure 14: More visualization results based on the S3DIS dataset in the 2-way 1-shot setting. The predictions for $p^{TxT}$ and $p^{DD}$.

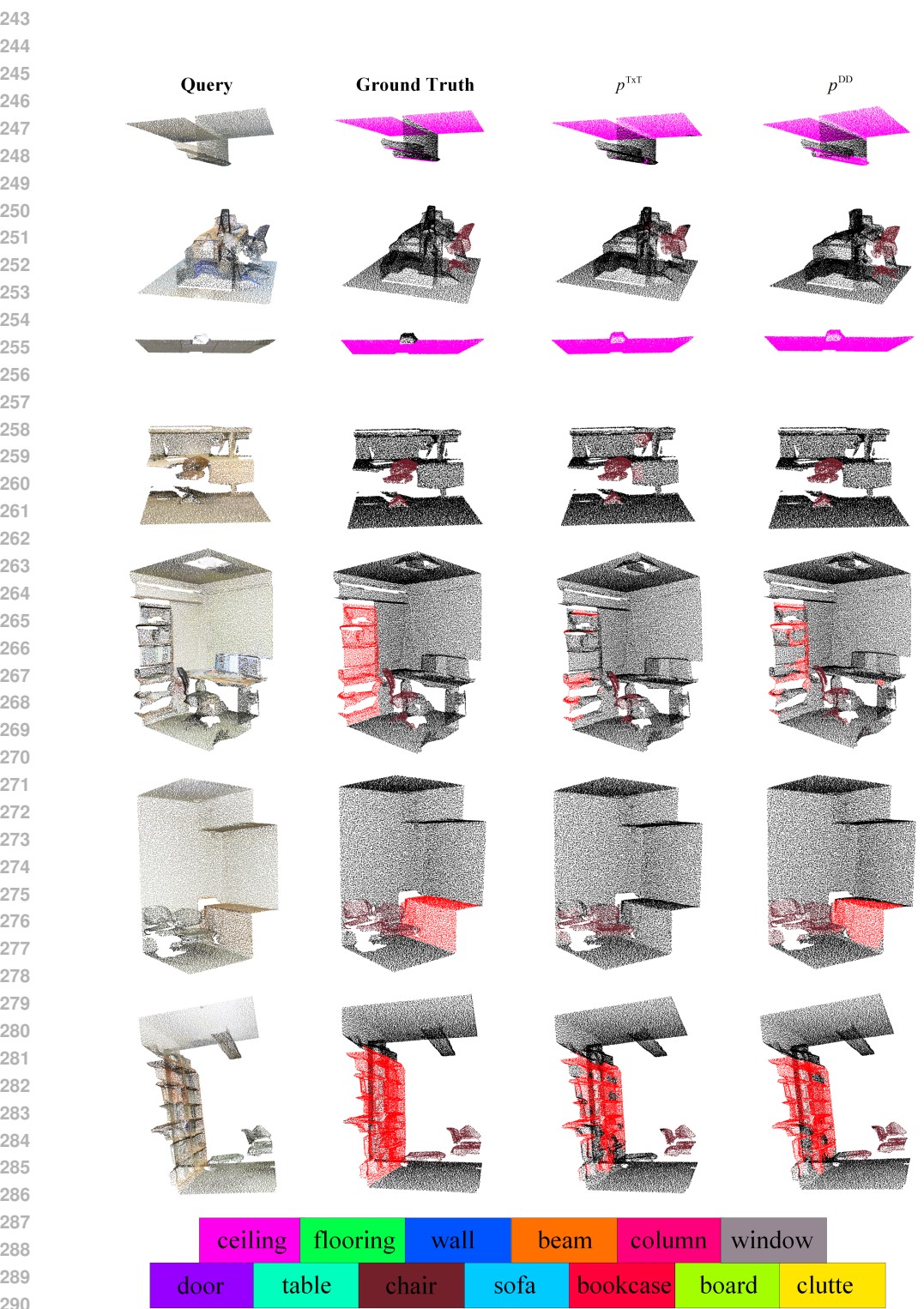

Figure 15: More visualization results based on the S3DIS dataset in the 2-way 1-shot setting. The predictions for the generator, average, and sum.

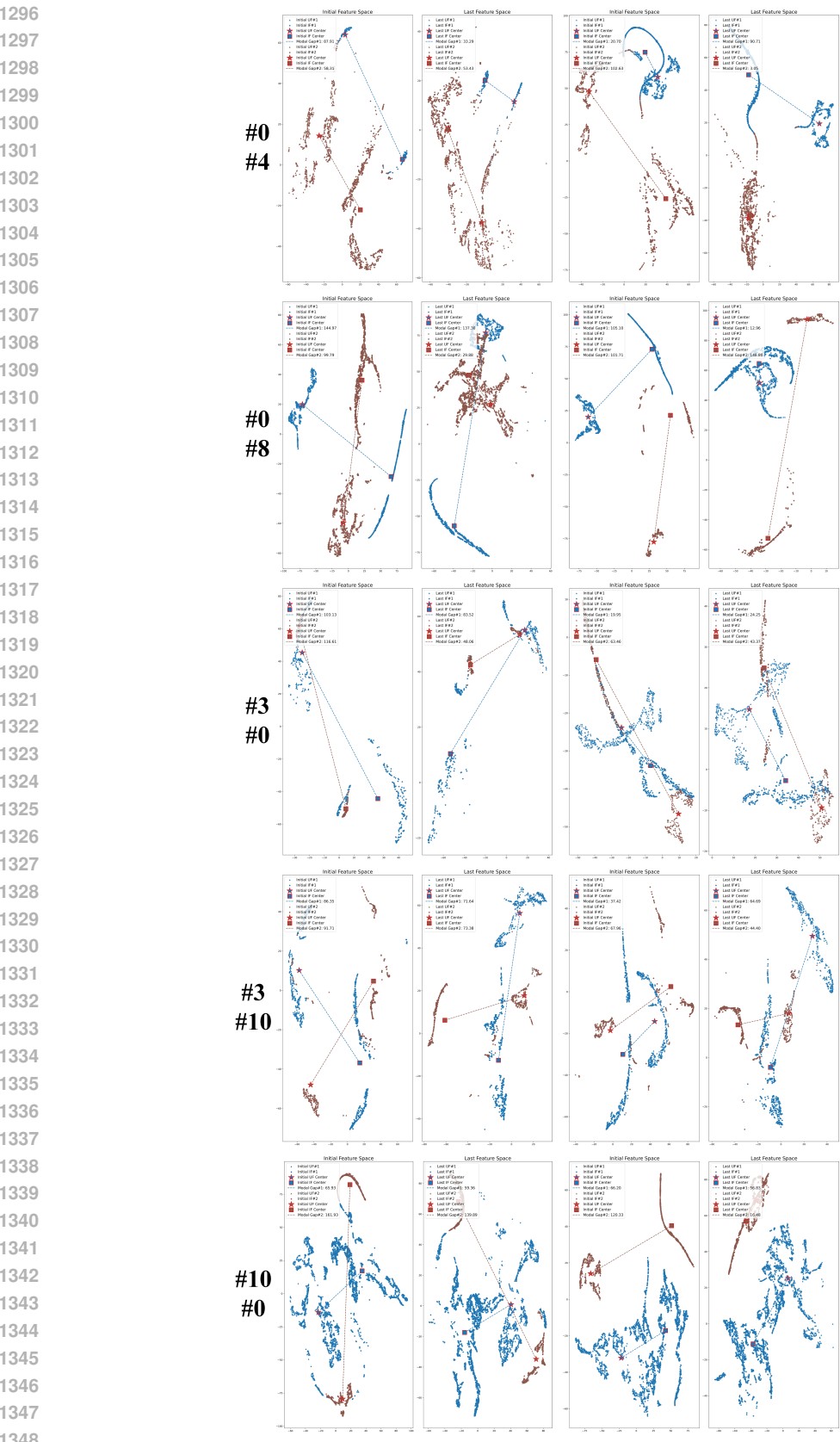

Figure 16: More visualization results of the Modal Gap under the S3DIS dataset in the 2-way 1-shot setting. We use t-SNE to project high-dimensional features onto a 2D display, visualizing the cluster centers for each category and the gaps between them.