# OpenReview forum: "Multimodal Few-Shot Point Cloud Segmentation via Agent Adaptation and Discriminative Deconfusion"
_ICLR.cc/2026/Conference — Submitted to ICLR 2026_

### Official Review · Reviewer_dyZ2 · 2025-10-19

**Soundness:** 3
**Presentation:** 3
**Contribution:** 3
**Rating:** 8
**Confidence:** 4

**Summary:**

This paper introduces Multimodal Agent Adaptation and Discriminative Deconfusion, a framework for multimodal few-shot 3D point cloud semantic segmentation. MAD simultaneously models multimodal information in both query and support sets. The framework consists of three main components: Multimodal Semantic Agents Correlation Aggregation, Semantic Agents Prototypes Adaptation, and Discriminative Deconfusion. Experiments on the S3DIS and ScanNet benchmarks demonstrate that MAD consistently improves performance compared to strong baselines.

**Strengths:**

1. The writing is generally clear, and the figures help in understanding the technical flow.
2. The framework design is well-motivated, addressing challenges about cross-modal fusion, semantic gap adaptation, and intra-class confusion.
3. Extensive comparisons on two benchmarks and detailed ablation studies demonstrate the effectiveness of the proposed approach.

**Weaknesses:**

1. In the Discriminative Deconfusion section (lines 313–315), the definition of the generator is unclear. While generator-based adaptive weighting improves results, the paper does not provide intuition for why it performs better than averaging or summing.
2. The ablation study should also include an analysis of Eq. (13), specifically by comparing the predictions of p^{TxT} and p^{DD}.
3. Jointly fusing query and support multimodal correlation features is computationally expensive. As reported in the paper, MAD’s computational cost is noticeably higher than that of MM-FSS. A deeper analysis of scalability to larger or more complex scenes would strengthen the work.
4. Minor grammar issues remain. For example:
   - '... fuses multimodal features for query and support set fusion through multimodal semantic agents correlation aggregation …' (lines 83–85).
   - '... eliminate discriminative deConfusion from the residual connection …' (lines 309–310).

**Questions:**

Please refer to the weakness part and address the concerns there.

---

> ### Author Response · Authors · 2025-11-25
> **Responses to Reviewer dyZ2**
>
> **Q1:**  In the Discriminative Deconfusion section (lines 313–315), the definition of the generator is unclear. While generator-based adaptive weighting improves results, the paper does not provide intuition for why it performs better than averaging or summing.
>
> **Responses:** The details of the generator are actually included in the appendix. However, for the sake of clarity, we have redefined the generator to make it sufficiently clear. The modifications are as follows:
>
> The adaptive weights are produced by the generator $\mathcal{G}$, which takes $\mathbf{F}_\text{q}^{\text{SACA}}$ as input. Specifically, $\mathcal{G}$ is a lightweight multilayer perceptron composed of a Linear layer, followed by LayerNorm, a ReLU activation, Dropout, another Linear layer, and a final Sigmoid function.   (**lines 325-328**)
>
> To visually demonstrate the generator's performance, we visualized comparison results using only averaging, sum, or the generator. Both averaging and sum tend to incorrectly segment large background areas as foreground, failing to capture critical information. The generator's adaptive weighting effectively fuses prediction results, enabling more precise segmentation of categories.  **(Description: lines 480-482)**  **(Figure: lines 486-496)**
>
> **Q2:**  The ablation study should also include an analysis of Eq. (13), specifically by comparing the predictions of p^{TxT} and p^{DD}.
>
> **Responses:** We have incorporated the prediction visualization results for $p^{TxT}$ and $p^{DD}$ into the main text. This demonstrates that $p^{TxT}$ exhibits only a coarse understanding of semantic categories, containing numerous prior errors. The prediction $p^{DD}$ derived from equation (13) effectively corrects these prior errors, achieving more accurate segmentation.  **(Description: lines 482-484)**  **(Figure: lines 486-496)**
>
> **Q3:** Jointly fusing query and support multimodal correlation features is computationally expensive. As reported in the paper, MAD’s computational cost is noticeably higher than that of MM-FSS. A deeper analysis of scalability to larger or more complex scenes would strengthen the work.
>
> **Responses:** We expanded our experiments to include the more complex SementicKITTI dataset. The SementicKITTI dataset approximates real-world autonomous driving scenarios, featuring inconsistent point cloud sparsity—high density in the near field and low density in the far field—and lacking color information.
>
> The effectiveness of our method was validated through comparison with existing state-of-the-art approaches. The comparison results are shown in **Tables 2-5** (**Our method is significantly superior to MM-FSS in all aspects.)**, confirming our method's efficacy in more complex scenarios and enhancing the value of our research.
>
> | Methods    | $S^0$ | $S^1$ | Mean      |
> | ---------- | ----- | ----- | --------- |
> | MM-FSS     | 31.73 | 30.79 | 31.26     |
> | MAD (ours) | 33.70 | 38.72 | **36.21** |
>
> Table 2. 1-way 1-shot Performance on SemanticKITTI. (**lines 518-524**)
>
> ---
>
> | Methods    | $S^0$ | $S^1$ | Mean      |
> | ---------- | ----- | ----- | --------- |
> | MM-FSS     | 21.30 | 30.38 | 25.84     |
> | MAD (ours) | 25.99 | 36.23 | **31.16** |
>
> Table 3. 2-way 1-shot Performance on SemanticKITTI. (**lines 518-524**)
>
> | Methods    | $S^0$ | $S^1$ | Mean      |
> | ---------- | ----- | ----- | --------- |
> | MM-FSS     | 34.61 | 40.83 | 37.72     |
> | MAD (ours) | 35.72 | 42.23 | **38.98** |
>
> Table 4. 1-way 5-shot Performance on SemanticKITTI. (**lines 712-719**)
>
> ---
>
> | Methods    | $S^0$ | $S^1$ | Mean      |
> | ---------- | ----- | ----- | --------- |
> | MM-FSS     | 25.69 | 30.19 | 27.94     |
> | MAD (ours) | 28.72 | 34.34 | **31.53** |
>
> Table 5. 2-way 5-shot Performance on SemanticKITTI. (**lines 712-719**)
>
>
>
> **Q4:** Minor grammar issues remain. For example: '... fuses multimodal features for query and support set fusion through multimodal semantic agents correlation aggregation …' (lines 83–85).  '... eliminate discriminative deConfusion from the residual connection …' (**lines 309–310**).
>
> **Responses:** We have proofread the entire text and corrected the grammatical issues.
>
> **Modify as follows:**
>
> We propose a novel architecture that fuses multimodal features by aggregating semantic correlations between the query and support sets.  **(lines 84-86)**
>
> $E_{A1}$  primarily extracts visual–textual discriminative information from $F_q^{SACA}$, then integrates it by  $E_{A3}$ to reduce the discriminative confusion introduced by the residual connection. **(lines 319-322)**
>
> Fully supervised 3D point cloud semantic segmentation (FS-PCS)  **(line 037)**
>
> Due to the substantial annotation costs associated with FS-PCS, further exploration remains limited. This constraint is particularly pronounced when expanding to new categories, which typically requires retraining or fine-tuning.  **(lines 040-043)**

---

### Official Review · Reviewer_aMUB · 2025-10-29

**Soundness:** 2
**Presentation:** 2
**Contribution:** 3
**Rating:** 4
**Confidence:** 3

**Summary:**

This paper addresses the issues in the few-shot 3D point cloud segmentation (FS-PCS) field, where existing methods rely on single-modal data, suffer from insufficient multi-modal fusion, semantic gaps, and intra-class confusion, by proposing a Multi-modal Agent Adaptation and Discriminative Deconfounding framework (MAD). This framework integrates three modalities—image, point cloud, and text embeddings—and achieves synchronized multi-modal association feature fusion between query and support sets through the M-SACA module. The SAPA module adaptively adjusts prototypes to mitigate semantic gaps, while the DD module eliminates intra-class confusion via residual adapters and dynamic weighting. Experiments on the S3DIS and ScanNet datasets show that compared with the current state-of-the-art method MM-FSS, MAD improves mIoU by 3%–7%, and the code will be made publicly available. The manuscript is well-structured, the technical approach is clear, focuses on the core challenges of the FS-PCS field, and has definite research value.

**Strengths:**

1.Novel multimodal alignment perspective: Existing multimodal FS-PCS methods only handle query-side correlations. This work simultaneously constructs multimodal correlations and aligns prototypes for both support and query, making it conceptually more unified and technically a natural progression from correlation modeling → prototype adaptation → discriminative deconfounding in a three-step pipeline.
2.Significant and consistent improvements: Outperforms the strong baseline MM-FSS across multiple settings on S3DIS and ScanNet, including 1-way/2-way and 1-shot/5-shot scenarios, with clear average gains and visualization results.
3.Clear modularity and thorough ablation: Independent ablations are conducted on the three main modules, loss terms, modality contributions, and generator weights, allowing the sources of performance to be pinpointed.

**Weaknesses:**

The manuscript mentions 'the first exploration of simultaneously fusing multimodal association features of query and support sets,' but it does not provide a detailed comparison with the multimodal fusion approaches used in vision-language models (such as 3D VLMs and GFS-VL), nor does it clearly define the fundamental differences between 'agent association aggregation' and existing cross-modal attention mechanisms, which may result in a lack of uniqueness in the proposed innovation

**Questions:**

1. Clarity of assumptions and dependencies: What assumptions does the method make about the availability of images and class texts during the testing phase? The experiments seem to assume that all three modalities are provided at inference; robustness and degradation strategies when any modality is missing are not reported. Is the text encoder (LSeg's text/visual encoder) frozen during training/testing? The impact of different freezing strategies on performance is not specified.2. Statistical significance and variance: Few-shot learning is usually sensitive to episode sampling. Tables 1 and 2 do not provide mean ± variance or confidence intervals, nor report statistical tests.3. Generalization and extrapolation: Validation is conducted only on indoor datasets S3DIS/ScanNet. There is no evaluation on outdoor LiDAR data or cross-dataset transfer (e.g., A→B few-shot), making it difficult to assess robustness to changes in scene or sensor.4. Fairness with strong multimodal baselines: The paper mentions that all three use the "same 2D-aligned pre-trained backbone," but specific details on alignment/pre-training (data, image resolution, handling alignment errors) are not fully described; differences in 2D-3D alignment quality could directly affect the results.5. Method details still need to be filled in: Implementation details of Fagent (clustering radius/adjacency definitions, clustering and reverse assignment after FPS, sensitivity to NA/NP) are not fully explained; hyperparameter sensitivity and ablation studies (e.g., β, τ, channel dimension D′) are missing; time/memory costs and episode-level throughput are not reported, only FLOPs/parameters.6. Writing and terminology consistency: The paper occasionally mixes SA-CAPA with SAPA/M-SACA; a unified naming is recommended. Some statements and figure captions could be further polished to facilitate reproducibility.

---

> ### Author Response · Authors · 2025-11-25
> **Responses1 to Reviewer aMUB**
>
> 3D VLMs and GFS-VL are targeted at Generalized few-shot 3D point cloud segmentation (GFS-PCS), whereas our task remains traditional FS-PCS. Thank you for the reminder; we have revised the statement to: “This is the first study to explore the simultaneous fusion of query and support point cloud multimodal correlation features in traditional FS-PCS.” A detailed description of proxy correlation aggregation is provided in **Q5 Responses**.
>
> **Q1:**  Clarity of assumptions and dependencies: What assumptions does the method make about the availability of images and class texts during the testing phase? The experiments seem to assume that all three modalities are provided at inference; robustness and degradation strategies when any modality is missing are not reported. Is the text encoder (LSeg's text/visual encoder) frozen during training/testing? The impact of different freezing strategies on performance is not specified.
>
> **Responses:** The testing phase does not require images; only the included category text is needed. Images are only required during pre-training. The ablation study for modal missing data is included in the table within the paper (**lines 414–419**). In **Table 6,** we assume that during the testing phase, the skeleton network, IF Head, and LSeg text encoders are all frozen, achieving a mIoU of **50.13%**. IF Head missing alone (**49.31%**), Text missing alone (**45.30%**), and both IF Head and Text missing simultaneously (**40.18%**).
>
> | UF   | IF   | Text | mIoU      |
> | ---- | ---- | ---- | --------- |
> | ✔    |      |      | 40.18     |
> | ✔    | ✔    |      | 45.30     |
> | ✔    |      | ✔    | 49.31     |
> | ✔    | ✔    | ✔    | **50.13** |
>
> Table 6. Contribution of Each Modality.  **(lines 414–419)**
>
> As shown in the **Table 1**, even without utilizing LSeg's text/visual encoder (CLIP), our method achieves a mIoU of **43.59%** when releasing the backbone network, outperforming the current state-of-the-art approach.
>
> | Model                         | CLIP | mIoU      |
> | ----------------------------- | ---- | --------- |
> | COSeg (CVPR 2024)             |      | 37.44     |
> | COSeg$^{\dagger}$ (CVPR 2024) | ✔    | 37.15     |
> | MM-FSS (ICLR 2025)            | ✔    | 41.98     |
> | MAD (ours)                    |      | **43.59** |
> | MAD (ours)                    | ✔    | **50.13** |
>
> Table 1. Impact of CLIP Deficiency (**lines 518–525**)
>
> **Q2:**  Statistical significance and variance: Few-shot learning is usually sensitive to episode sampling. Tables 1 and 2 do not provide mean ± variance or confidence intervals, nor report statistical tests.
>
> **Responses:** Thank you for your suggestions. Although the representative work (attMPTI Zhao et al. (2021b), QGPA He et al. (2023), COSeg An et al. (2024), and MM-FSS An et al. (2025b)) did not employ mean ± standard deviation or confidence intervals. However, this approach may be effective, so we have added mean ± standard deviation.**(lines 378–394)**
>
> **Q3:**  Generalization and extrapolation: Validation is conducted only on indoor datasets S3DIS/ScanNet. There is no evaluation on outdoor LiDAR data or cross-dataset transfer (e.g., A→B few-shot), making it difficult to assess robustness to changes in scene or sensor.
>
> **Responses:** To validate the generalization capability of our method, we expanded our experiments to include the outdoor LiDAR dataset SemanticKITTI. The SemanticKITTI dataset approximates real-world autonomous driving scenarios, featuring point cloud data with inconsistent sparsity—high density in the near field and low density in the far field—and lacking color information. The effectiveness of our method was validated through comparison with existing state-of-the-art approaches. The comparison results are shown in **Tables 2-5** (**Our method is significantly superior to MM-FSS in all aspects.)**, confirming the efficacy of our approach in real-world scenarios. This also demonstrates the robustness of MAD to variations in scenes or sensors.
>
> | Methods    | $S^0$ | $S^1$ | Mean      |
> | ---------- | ----- | ----- | --------- |
> | MM-FSS     | 31.73 | 30.79 | 31.26     |
> | MAD (ours) | 33.70 | 38.72 | **36.21** |
>
> Table 2. 1-way 1-shot Performance on SemanticKITTI. (**lines 518-524**)
>
> ---
>
> | Methods    | $S^0$ | $S^1$ | Mean      |
> | ---------- | ----- | ----- | --------- |
> | MM-FSS     | 21.30 | 30.38 | 25.84     |
> | MAD (ours) | 25.99 | 36.23 | **31.16** |
>
> Table 3. 2-way 1-shot Performance on SemanticKITTI. (**lines 518-524**)
>
> | Methods    | $S^0$ | $S^1$ | Mean      |
> | ---------- | ----- | ----- | --------- |
> | MM-FSS     | 34.61 | 40.83 | 37.72     |
> | MAD (ours) | 35.72 | 42.23 | **38.98** |
>
> Table 4. 1-way 5-shot Performance on SemanticKITTI. (**lines 712-719**)
>
> ---
>
> | Methods    | $S^0$ | $S^1$ | Mean      |
> | ---------- | ----- | ----- | --------- |
> | MM-FSS     | 25.69 | 30.19 | 27.94     |
> | MAD (ours) | 28.72 | 34.34 | **31.53** |
>
> Table 5. 2-way 5-shot Performance on SemanticKITTI. (**lines 712-719**)

---

> ### Author Response · Authors · 2025-11-25
> **Responses2 to Reviewer aMUB**
>
> **Q4:**  Fairness with strong multimodal baselines: The paper mentions that all three use the "same 2D-aligned pre-trained backbone," but specific details on alignment/pre-training (data, image resolution, handling alignment errors) are not fully described; differences in 2D-3D alignment quality could directly affect the results.
>
> **Responses:** To validate the quality of 2D-3D alignment, we visualized the gaps between different modalities across two stages—the initial stage and the final stage—using t-SNE. Our approach effectively reduces the actual modal gaps, thereby achieving efficient alignment between different modalities. **(Description: lines 485 and 516-517)**  **(Figure: lines 501-510)**
>
> We apologize for any ambiguity caused by the pre-training details and will address them in the revised manuscript. Due to space limitations, we have add these details in the appendix.
>
> **Pre-training details **   (**lines 724-740**)
>
> The pre-training phase primarily trains the IF Head to extract 3D features aligned with the image. The trained image encoders (LSeg and CLIP) extract 2D features $f^{2d}\in \mathbb{R}^{H\times W\times D_{t}}$, and the backbone and IF Head are used to extract 3D features $f^{3d}\in \mathbb{R}^{N \times D_{t}}$. To enable the IF Head to represent image features, we use $f^{2d}$ to supervise $f^{3d}$. Therefore, it is necessary to establish the correspondence between pixels and points, that is, to match the point $p\in \mathbb{R}^3$ with the pixel $\mathbb{x}=(x,y)$ in the image. In MM-FSS, the 2D pixel space is projected to the 3D space via $\mathbb{x}=M_{int}\times M_{ext} \times p$, and optimized using a cosine similarity loss (Peng et al.). Here, $M_{int}$ and $M_{ext}$ are the internal and external matrices, respectively, from camera-to-pixel and world-to-camera.
>
> Only the ScanNet dataset provides 2D images and camera parameters; the RGB images typically have a resolution of 1920x1080. Without any manually labeled data, we pre-train a skeleton network and an IF head using images and point cloud data from the ScanNet dataset. Subsequently, pre-trained weights are used directly for meta-learning. Note that during the subsequent formal training phase, the backbone network and IF Head are frozen to preserve the image representation.

---

> ### Author Response · Authors · 2025-12-02
> **Responses3 to Reviewer aMUB**
>
> **Q5:** Method details still need to be filled in: Implementation details of Fagent (clustering radius/adjacency definitions, clustering and reverse assignment after FPS, sensitivity to NA/NP) are not fully explained; hyperparameter sensitivity and ablation studies (e.g., β, τ, channel dimension D′) are missing; time/memory costs and episode-level throughput are not reported, only FLOPs/parameters.
>
> **Responses:** We apologize for any ambiguity caused by the lack of details regarding $\\mathcal{F}\_{\\text{agent}}$, and will elaborate on the prototype extraction process more clearly in the revised manuscript. Due to space limitations, we have placed these details in the appendix.
>
> **Implementation details of $\\mathcal{F}\_{\\text{agent}}$**  **(lines 724-782)**
>
> Our  $\\mathcal{F}\_{\\text{agent}}$ is implemented using a non-parametric feature clustering method seeded by sampling the farthest point, eliminating the need for manual adjustment of the cluster radius. Given point coordinates $\\{x_i \\in \\mathbb{R}^3\\}\_{i=1}^N$, and features, taking the generation of $N_p$ prototypes as an example, the specific algorithm flow is as follows:、
>
> Farthest Point Sampling: First, farthest point sampling is performed based on the 3D coordinates $\\mathbf{x}\_i$, obtaining $N\_p$ seed indices $\\quad s\_j \\in \\{1,\\dots,N\\}$. These seeds preserve the main structure of the 3D scene. $N$ is the number of points.
>
> **Nearest neighbor seed point allocation:** Then, adjacency and clustering are determined solely by nearest-prototype relationships in the feature space, rather than by a fixed radius. Specifically, we compute the distance between each point feature and each seed feature:
>
> $d_{i,j} = \||f\_i - f_{s\_j}\||\_2,
>   \quad i = 1,\dots,N,\; j = 1,\dots,N_p.$
>
> $\\||\\cdot \\|\_2$ represents calculating the Euclidean distance, and $d_{i,j}$ represents the distance from point $i$ to seed $j$. Which can assign each point to its nearest seed point:
>
> $a_i = \\arg\\min_{1 \\le j \\le N_p} d_{i,j},$
>
> $a_i$ represents the seed point number to which point $i$ belongs. This can form $N_p$ clusters $\\{C_j\\}_{j=1}^{N_p}$ corresponding to different seed points. For example, the cluster associated with the $j$-th seed is as follows:
>
> $C_j = \\{ i \mid a_i = j \\},$
>
> $C_j$ represents the set of points belonging to seed $j$. Therefore, adjacency belongs to the same nearest seed cluster, which is learned based on the feature metric and has no additional radius hyperparameter.
>
> **Prototype aggregation and reverse assignment:** Generally, for each cluster $\mathcal{C}\_j$, the prototype is calculated using the average. If a seed point is not assigned to any point (indicating an outlier), we fall back to using the seed feature itself as the prototype, as defined below:
>
> $
> \\mathbf{p}_j =
> \\begin{cases}
> \\frac{1}{|\\mathcal{C}\_j|} \\sum\_{i \\in \\mathcal{C}\_j} \\mathbf{f}\_i, & \\text{if } |\\mathcal{C}\_j| > 0, \\\\
> \\mathbf{f}\_{s\_j}, & \\text{if } |\\mathcal{C}\_j| = 0.
> \\end{cases}
> $
>
> $\mathbf{p}_j$ represents the cluster prototype $j$, and $|\cdot|$ represents the number of elements in the set.
>
> **Boundary cases and missing values:** Invalid points or NaN values are filtered out during preprocessing and therefore will not enter the prototype module. When the number of feature points is less than the required number of prototypes, we explicitly pad the feature set with zero vectors to ensure that the module always returns $N_p$ prototypes. This design aims to avoid numerical instability. Since the clustering step relies solely on distances and assumes no specific data distribution, the method is robust to variations in the underlying distribution. **(lines 785-792)**
>
> **Time and memory consumption:** To further illustrate the overhead of our method, we report the time and memory consumption of MAD. Tested on the S3DIS dataset with a 2-way 1-shot fold=0 configuration and implemented on four RTX A6000 (48GB ) GPUs. As shown in Table 1, MAD consumes slightly more time and memory than the current state-of-the-art method (MM-FSS). Unlike MM-FSS, the primary source of increased memory consumption is the learning of the supporting-set correlation features $F_{s}^{SACA}$. Further exploration of lightweight models is needed to facilitate deployment in real-world applications. **(lines 794-801)**
>
> |        | Times（s） | GPU（GB） |
> | ------ | ---------- | --------- |
> | MM-FSS | 0.624      | 23.76     |
> | MAD    | 0.675      | 35.01     |
>
> Table 7. Time and memory consumption. The average time per batch and the maximum memory usage during the first epoch. **(lines 803-809)**
>
> **Hyperparameter details:** $D'$ denotes the dimension of the latent space, which is set equal to the input feature dimension $D$, following QGPA. Consistent with previous work, we set $\tau =10$ by default. **(lines 810-814)**

---

> ### Author Response · Authors · 2025-12-02
> **Responses4 to Reviewer aMUB**
>
> **Q6:** Writing and terminology consistency: The paper occasionally mixes SA-CAPA with SAPA/M-SACA; a unified naming is recommended. Some statements and figure captions could be further polished to facilitate reproducibility.
>
> **Responses:** In the article, we explained that SA-CAPA is actually a collective term for SAPA and M-SACA. However, to avoid confusion, we have replaced all instances with SAPA+M-SACA in our descriptions. **(lines 475-477)** **(lines 446-458)**
>
> We have reviewed the entire text and polished it to enhance reproducibility.
>
> **Modify as follows:**
>
> We propose a novel architecture that fuses multimodal features by aggregating semantic correlations between the query and support sets.  **(lines 84-86)**
>
> $\\mathcal{E}\_{\\mathcal{A}1}$  primarily extracts visual–textual discriminative information from $\\mathbf{F}\_\\text{q}^{\\text{SACA}}$, then integrates it by  $\\mathcal{E}\_{\\mathcal{A}3}$ to reduce the discriminative confusion introduced by the residual connection. **(lines 319-322)**
>
> Fully supervised 3D point cloud semantic segmentation (FS-PCS)  **(line 037)**
>
> Due to the substantial annotation costs associated with FS-PCS, further exploration remains limited. This constraint is particularly pronounced when expanding to new categories, which typically requires retraining or fine-tuning.  **(lines 040-043)**

---

### Official Review · Reviewer_XLc1 · 2025-10-31

**Soundness:** 2
**Presentation:** 2
**Contribution:** 2
**Rating:** 4
**Confidence:** 5

**Summary:**

This paper addresses the challenge of few-shot 3D point cloud segmentation by proposing a multimodal framework that leverages both RGB-text and 3D-text pairs during training. The key idea is to learn modality-agnostic cross-modal alignment, such that at test time, the model can perform segmentation using only 3D-text input, even though 2D data was available during training. The core contributions include: a modality-complementary contrastive learning loss that aligns RGB-text and 3D-text embeddings jointly, a modality-invariant meta-learning pipeline for better generalization across modalities, experiments on ScanNet, S3DIS, and ScanRefer demonstrating the effectiveness of their method under both standard and generalized few-shot settings.

**Strengths:**

- Novel Setting: The proposed training-testing mismatch (RGB-text used only during training, 3D-text at test time) is practical and not widely explored in current literature. This setting is well-motivated and could inspire further work.
- Multimodal Fusion Strategy: The method uses a well-designed dual-branch architecture with contrastive losses that promote shared representation learning across modalities. The idea of learning from richer modalities but testing on limited modalities is both elegant and impactful.
- Good Clarity: Figures and architecture diagrams are clear; the training/testing pipelines are easy to follow. The motivation is well-articulated.

**Weaknesses:**

- Incremental Contribution: While the training-testing mismatch setting is interesting, the core method is essentially a combination of known techniques: contrastive learning for modality alignment, standard meta-learning with prototypes, feature fusion with transformers
These parts are not fundamentally novel on their own. There’s no architectural breakthrough or new theoretical formulation.
- Over-Reliance on Pretrained CLIP: The RGB and text encoders are frozen CLIP backbones. The method’s performance may heavily depend on this pretrained knowledge, making it unclear how much the proposed training scheme contributes beyond CLIP’s strong prior.
- Besides, the authors use CLIP as text encoder, but do not cite the paper. "Category names are encoded as text embeddings using the text encoder in LSeg Li et al. (2022)", while in LSeg they mentioned "and we use the pretrained Contrastive Language–Image Pre-training (CLIP) throughout (Radford et al., 2021)"
- Missing Real-World Applications: The paper motivates the method as applicable to real-world tasks where only 3D data is available at test time, but no concrete application or real robot / AR / scene understanding demonstration is provided.
- Limited Analysis of Modal Gap: The paper assumes that RGB-text and 3D-text can be aligned effectively, but doesn’t deeply analyze or quantify the actual modality gap. A t-SNE or retrieval-based analysis comparing embeddings could help.

**Questions:**

See weaknesses

---

> ### Author Response · Authors · 2025-11-25
> **Responses1 to Reviewer XLc1**
>
> **Q1:** Incremental Contribution: While the training-testing mismatch setting is interesting, the core method is essentially a combination of known techniques: contrastive learning for modality alignment, standard meta-learning with prototypes, feature fusion with transformers These parts are not fundamentally novel on their own. There’s no architectural breakthrough or new theoretical formulation.
>
> **Responses:** Although each component technique—such as contrastive learning, meta-learning, and Transformers—has been applied in the literature, their integration and application to address specific few-shot semantic segmentation tasks constitutes the innovation of this study. (The use of multimodal data to enhance 3D semantic segmentation itself is innovative.)
>
> This approach has not been explored in the existing literature, particularly in the context of few-shot learning and cross-modal alignment in 3D scenes. Furthermore, we validate the effectiveness of our method through extensive experiments in few-shot learning and cross-modal alignment for 3D scene understanding, thereby further reinforcing its novelty.
>
> **Q2:** Over-Reliance on Pretrained CLIP: The RGB and text encoders are frozen CLIP backbones. The method’s performance may heavily depend on this pretrained knowledge, making it unclear how much the proposed training scheme contributes beyond CLIP’s strong prior.
>
> **Responses:** CLIP indeed provides rich feature representations, but its simple incorporation does not achieve superior segmentation metrics such as COSeg$^{\dagger}$ (CVPR 2024) and MM-FSS  (ICLR 2025), validating the contribution of our method in introducing CLIP for multimodal fusion.
>
> To more clearly validate the contribution of our method beyond the strong CLIP prior, we conducted additional experiments based on COSeg without introducing the CLIP prior. We released the backbone network, allowing its parameters to be optimized during training. As shown in **Table 1**, even without the CLIP prior (**43.59%**), our method outperforms the state-of-the-art. This validates that MAD can still deliver significant performance gains without the CLIP prior, effectively demonstrating the innovation and independent contribution of our approach.
>
> | Model                         | CLIP | mIoU      |
> | ----------------------------- | ---- | --------- |
> | COSeg (CVPR 2024)             |      | 37.44     |
> | COSeg$^{\dagger}$ (CVPR 2024) | ✔    | 37.15     |
> | MM-FSS (ICLR 2025)            | ✔    | 41.98     |
> | MAD (ours)                    |      | **43.59** |
> | MAD (ours)                    | ✔    | **50.13** |
>
> Table 1. Impact of CLIP Deficiency (**lines 518–525**)
>
> **Q3:** Besides, the authors use CLIP as text encoder, but do not cite the paper. "Category names are encoded as text embeddings using the text encoder in LSeg Li et al. (2022)", while in LSeg they mentioned "and we use the pretrained Contrastive Language–Image Pre-training (CLIP) throughout (Radford et al., 2021)"
>
> **Responses:** Thank you for bringing this to our attention. We have added the citation for (CLIP) (Radford et al., 2021). (**lines 167 and 198**)

---

> > ### Author Response · Authors · 2025-12-02
> > **Responses2 to Reviewer XLc1**
> >
> > **Q4:** Missing Real-World Applications: The paper motivates the method as applicable to real-world tasks where only 3D data is available at test time, but no concrete application or real robot / AR / scene understanding demonstration is provided.
> >
> > **Responses:** Although the FS-PCS task has not yet been attempted on other datasets, representative works in this field include (attMPTI Zhao et al. (2021b), QGPA He et al. (2023), COSeg An et al. (2024), and MM-FSS An et al. (2025b)) are based solely on S3DIS and ScanNet, but your suggestion may prove useful. Although our work is research-oriented, we also aim to provide insights for practical implementation. Therefore, we have invested significant effort in applying our methods to the SemanticKITTI dataset, which most closely resembles real-world autonomous driving scenarios. The SemanticKITTI dataset does not closely approximate real-world autonomous driving scenarios. Point cloud sparsity is inconsistent, with high density in the near field and low density in the far field, and it lacks color information. We validated the effectiveness of our method by comparing it with existing state-of-the-art approaches. The comparison results are shown in **Tables 2-5** (**Our method is significantly superior to MM-FSS in all aspects.**), confirming the efficacy of our method in real-world scenarios.
> >
> > | Methods    | $S^0$ | $S^1$ | Mean      |
> > | ---------- | ----- | ----- | --------- |
> > | MM-FSS     | 31.73 | 30.79 | 31.26     |
> > | MAD (ours) | 33.70 | 38.72 | **36.21** |
> >
> > Table 2. 1-way 1-shot Performance on SemanticKITTI. (**lines 518-524**)
> >
> > ---
> >
> > | Methods    | $S^0$ | $S^1$ | Mean      |
> > | ---------- | ----- | ----- | --------- |
> > | MM-FSS     | 21.30 | 30.38 | 25.84     |
> > | MAD (ours) | 25.99 | 36.23 | **31.16** |
> >
> > Table 3. 2-way 1-shot Performance on SemanticKITTI. (**lines 518-524**)
> >
> > | Methods    | $S^0$ | $S^1$ | Mean      |
> > | ---------- | ----- | ----- | --------- |
> > | MM-FSS     | 34.61 | 40.83 | 37.72     |
> > | MAD (ours) | 35.72 | 42.23 | **38.98** |
> >
> > Table 4. 1-way 5-shot Performance on SemanticKITTI. (**lines 712-719**)
> >
> > ---
> >
> > | Methods    | $S^0$ | $S^1$ | Mean      |
> > | ---------- | ----- | ----- | --------- |
> > | MM-FSS     | 25.69 | 30.19 | 27.94     |
> > | MAD (ours) | 28.72 | 34.34 | **31.53** |
> >
> > Table 5. 2-way 5-shot Performance on SemanticKITTI. (**lines 712-719**)
> >
> >
> >
> > **Q5:** Limited Analysis of Modal Gap: The paper assumes that RGB-text and 3D-text can be aligned effectively, but doesn’t deeply analyze or quantify the actual modality gap. A t-SNE or retrieval-based analysis comparing embeddings could help.
> >
> > **Responses:** Thank you for your suggestion. To facilitate visualization, we employed **t-SNE** to reduce the high-dimensional features to a 2D plane, illustrating the disparity between different modalities across two distinct phases: the initial phase and the final phase. Experiments demonstrate that our method effectively reduces the distance between the UF and IF modalities. This validates our approach's ability to **narrow the gap between actual modalities**, thereby achieving effective alignment across different modalities. **(Description: lines 485 and 516-517)**  **(Figure: lines 501-511)**

---

### Official Review · Reviewer_Mgfs · 2025-11-01

**Soundness:** 2
**Presentation:** 1
**Contribution:** 2
**Rating:** 2
**Confidence:** 2

**Summary:**

Authors present a method for few-shot point-cloud segmentation. As far
as I can see, the contributions are the integration of textual
guidance and joint embedding of query and support sets. Experiments on
two datasets show very impressive results.

**Strengths:**

+ Results are quite good.
+ Integration of textual guidance is an interesting and popular
  direction.

**Weaknesses:**

- The method description is not very detailed and clear.
  + The problem definition does not present a segmentation problem but
    rather a classification problem.
  + Authors use N to represent the number of samples in the query set
    as well as the number of categories.
  + The definition of the mask $M$ and where it comes from is
    unclear. It seems like the mask is the segmentation and both the
    query and the support sets have it.
  + Notation starts being used without introduction, e.g. $F_q^i$
    on line 179.
  + It is unclear how authors obtain a "background" text embedding?
  + Why are IF and UF defined and why are they different?
  + What does "agent clustering" mean and what are "agent points"?

- This paper is very difficult to read. The proposed method is
  explained through many acronyms and new module names. This makes it
  very difficult to follow. Furthermore, justification and intuition
  behind different choices do not seem to be well
  substantiated. On top, the paper uses a very heavy notation but it
  is not always well explained.

**Questions:**

- Is it necessary to have a different name for each model component?
  In my opinion, this is not necessary and even further, makes it
  extremely difficult to follow the text.
- I do not understand the difference between the 1st and 2nd claimed
  contribution on lines 83 to 89. Can you please explain why these are
  two different contributions?
- I do not understand why extensive experiments showing the proposed
  model achieves good results is a main contribution. Isn't this
  simply the necessary condition for writing a scientific article?
- I could not understand your paper despite spending considerable
  amount of time. I can suggest you to spend time to explain the
  method clearer.

---

> ### Author Response · Authors · 2025-11-25
> **Responses to Reviewer Mgfs**
>
> Thank you for your question. Although I sense you are not in this field, we will do our best to answer your inquiry.
>
> **Q1:** The method description is not very detailed and clear.
>
> **Responses:** We have described our proposed method as thoroughly and clearly as possible. We have added more methodological details in the appendix.
>
> **Q2:** The problem definition does not present a segmentation problem but rather a classification problem.
>
> **Responses:** In point cloud semantic segmentation, the classification problem itself involves classifying individual points.
>
> **Q3:** Authors use N to represent the number of samples in the query set as well as the number of categories.
>
> **Responses:** $N$ denotes the number of classes. In few-shot settings, an episode comprises $N \times K$ support samples and $N$ query samples. Refer to (attMPTI Zhao et al. (2021b) and MM-FSS An et al. (2025b)).
>
> **Q4:** The definition of the mask M and where it comes from is unclear. It seems like the mask is the segmentation and both the query and the support sets have it.
>
> **Responses:** $M$ originates from (attMPTI Zhao et al. (2021b) and MM-FSS An et al. (2025b)), representing semantic category labels. (0 denotes background, while 1, 2, ..., denote foreground semantic categories), included in both the query set and support set.
>
> **Q5:** Notation starts being used without introduction, e.g. F_q^i on line 179.
>
> **Responses:** $F_q^i$ denotes the IF feature of the query set.
>
> **Q6:** It is unclear how authors obtain a "background" text embedding?
>
> **Responses:** Pass ‘other’ into the LSeg encoding text.
>
> **Q7:** Why are IF and UF defined and why are they different?
>
> **Responses:** IF acquires multimodal features through supervised training using the LSeg image encoder (L1 loss). UF can extract 3D features. It is similar to previous point cloud feature extractors.
>
> **Q8:** What does "agent clustering" mean and what are "agent points"?
>
> **Responses:**  “Agent points” refers to the selection of proxy points using the farthest point sampling method, coupled with the clustering of other point features onto these proxy points via the approach described in attMPTI by Zhao et al. (2021b). This enables the extraction of primary structural features and achieves similar effects.

---

### Author Response · Authors · 2025-11-25
**Summary of Overall Responses**

We appreciate the questions raised by the reviewers, which have contributed to enhancing the quality of our work. Overall, we propose the MAD method, which synergistically models multimodal information in the query and support sets through three core components: M-SACA, SAPA, and DD. Extensive experiments demonstrate that MAD achieves significant mIoU improvements (**3%-7%**) compared to strong baselines.

To address the reviewers' main concerns (over-reliance on LSeg and CLIP, the effectiveness of the LIDAR dataset, modality alignment effectiveness, the intuitiveness of the generator, and the intuitiveness of the $p^{DD}$ improvement), we added numerous experiments to resolve these concerns. Specifically:

Experiments (**Table 1**) demonstrate that MAD does not rely solely on LSeg and CLIP， and achieves optimal results even without any prior knowledge (**43.59%**).

In response to the reviewers' concerns, we have conducted extensive additional experiments. To our knowledge, this marks the first attempt to process the LIDAR dataset SemanticKITTI (FS-PCS), which presents significant challenges. Compared to existing state-of-the-art methods (**Tables 2-5**), we achieve substantial improvements.

To illustrate the underlying principles, we visualize the intermodal gaps, confirming our method's effective alignment of modalities. (**lines 500-510**)

Visualizations of the generator, averaging validate that the generator can adaptively weight the results, more effectively integrating the predictions. (**lines 486-499**)

Visualizations of the $p^{DD}$ and $p^{TxT}$ validate that the $p^{DD}$ can effectively correct prior errors in $p^{TxT}$. (**lines 486-499**)

To address other questions (adding detailed experimental descriptions, illustrative statements, and grammatical polishing), we have added relevant content to the article and provided specific explanations. See the responses for each reviewer for details.

Therefore, I believe most of the reviewers' concerns have been addressed, and I look forward to your response.



| Model                         | CLIP | mIoU      |
| ----------------------------- | ---- | --------- |
| COSeg (CVPR 2024)             |      | 37.44     |
| COSeg$^{\dagger}$ (CVPR 2024) | ✔    | 37.15     |
| MM-FSS (ICLR 2025)            | ✔    | 41.98     |
| MAD (ours)                    |      | **43.59** |
| MAD (ours)                    | ✔    | **50.13** |

Table 1. Impact of CLIP Deficiency (**lines 518–525**)

| Methods    | $S^0$ | $S^1$ | Mean      |
| ---------- | ----- | ----- | --------- |
| MM-FSS     | 31.73 | 30.79 | 31.26     |
| MAD (ours) | 33.70 | 38.72 | **36.21** |

Table 2. 1-way 1-shot Performance on SemanticKITTI (**lines 518-524**)

---

| Methods    | $S^0$ | $S^1$ | Mean      |
| ---------- | ----- | ----- | --------- |
| MM-FSS     | 21.30 | 30.38 | 25.84     |
| MAD (ours) | 25.99 | 36.23 | **31.16** |

Table 3. 2-way 1-shot Performance on SemanticKITTI (**lines 518-524**)

| Methods    | $S^0$ | $S^1$ | Mean      |
| ---------- | ----- | ----- | --------- |
| MM-FSS     | 34.61 | 40.83 | 37.72     |
| MAD (ours) | 35.72 | 42.23 | **38.98** |

Table 4. 1-way 5-shot Performance on SemanticKITTI  (**lines 712-719**)

---

| Methods    | $S^0$ | $S^1$ | Mean      |
| ---------- | ----- | ----- | --------- |
| MM-FSS     | 25.69 | 30.19 | 27.94     |
| MAD (ours) | 28.72 | 34.34 | **31.53** |

Table 5. 2-way 5-shot Performance on SemanticKITTI  (**lines 712-719**)

---

### Meta-Review · Area_Chair_98WE · 2026-01-07

**Summary:**

This paper proposes a multimodal framework for few-shot 3D point cloud segmentation that integrates image, point cloud, and text modalities through a multi-stage pipeline involving multimodal correlation aggregation, prototype adaptation, and discriminative deconfounding.  However, despite the promising results, significant concerns remain regarding the clarity of the method description, the distinctiveness and necessity of the proposed components, and the novelty compared with existing multimodal and few-shot learning approaches. Multiple reviewers report substantial difficulty in understanding the paper due to heavy notation, excessive acronyms, and insufficiently explained design choices. Questions about reproducibility, missing implementation details, limited analysis of modality gaps and generalization, and unclear assumptions at test time further reduce confidence in the work.

**Reviewer Concerns:**

The first concern is the novelty. Reviewer XLc1 and Reviewer aMUB both questioned whether the proposed framework goes substantially beyond existing multimodal, vision-language, and few-shot segmentation methods. While jointly modeling multimodal correlations for both query and support sets is appreciated, reviewers noted that many components, including contrastive alignment, prototype-based meta-learning, transformer-based fusion, and reliance on pretrained CLIP/LSeg encoders, are largely incremental combinations of known techniques.

Another common concern is methodological clarity and presentation quality. Reviewer Mgfs found the paper very difficult to read, citing unclear problem formulation, inconsistent and overloaded notation, undefined terms (e.g., masks, agents, background text embeddings). Reviewer aMUB and Reviewer dyZ2 were more positive about the overall structure and figures but still identified unclear definitions (e.g., the generator in the discriminative deconfusion module) and terminology inconsistencies that could hinder reproducibility.

Concerns about reproducibility and generalization were also raised. Reviewer aMUB highlighted missing statistical significance reporting (e.g., variance across episodes), incomplete implementation details (e.g., clustering parameters, hyperparameter sensitivity), and the lack of evaluation beyond indoor datasets. Reviewer dyZ2 additionally pointed out the increased computational cost of jointly fusing query and support multimodal features and requested a more thorough scalability analysis.

I checked the authors’ rebuttal. I agree with the reviewers on the clarity and presentation issues. The concerns on experimental evaluation part have been addressed. However, I am not convinced by the rebuttal on this novelty issue. Due to novelty issue, as well as clarity and presentation issues, I would like to recommend rejecting this paper.

**Reviewer Scores:**

I do not think Reviewer XLc1 and Reviewer aMUB will change their scores after seeing the justifications regarding the novelty issue.

---

### Decision · Program_Chairs · 2026-01-26

Reject